# Zero-sum Polymatrix Markov Games: Equilibrium Collapse and Efficient Computation of Nash Equilibria

**Fivos Kalogiannis**
Department of Computer Science
University of California, Irvine
Irvine, CA
fkalogia@uci.edu

**Ioannis Panageas**
Department of Computer Science
University of California, Irvine
Irvine, CA
ipanagea@ics.uci.edu

## Abstract

The works of (Daskalakis et al., 2009, 2022; Jin et al., 2022; Deng et al., 2023) indicate that computing Nash equilibria in multi-player Markov games is a computationally hard task. This fact raises the question of whether or not computational intractability can be circumvented if one focuses on specific classes of Markov games. One such example is two-player zero-sum Markov games, in which efficient ways to compute a Nash equilibrium are known. Inspired by zero-sum polymatrix normal-form games (Cai et al., 2016), we define a class of zero-sum multi-agent Markov games in which there are only pairwise interactions described by a graph that changes per state. For this class of Markov games, we show that an $\epsilon$-approximate Nash equilibrium can be found efficiently. To do so, we generalize the techniques of (Cai et al., 2016), by showing that the set of coarse-correlated equilibria collapses to the set of Nash equilibria. Afterwards, it is possible to use any algorithm in the literature that computes approximate coarse-correlated equilibria Markovian policies to get an approximate Nash equilibrium.

## 1 Introduction

Multi-agent reinforcement learning (MARL) is the discipline that is concerned with strategic interactions between agents who find themselves in a dynamically changing environment. Early aspects of MARL can be traced back to, as early as, the initial text on two-player zero-sum stochastic/Markov games (Shapley, 1953). Today, Markov games have been established as the theoretical framework for MARL (Littman, 1994). The connection between game theory and MARL has lead to several recent cornerstone results in benchmark domains in AI (Bowling et al., 2015; Brown and Sandholm, 2019, 2018; Brown et al., 2020; Silver et al., 2017; Moravčík et al., 2017; Perolat et al., 2022; Vinyals et al., 2019). The majority of the aforementioned breakthroughs relied on computing *Nash equilibria* (Nash, 1951) in a scalable and often decentralized manner. Although the theory of single agent reinforcement learning (RL) has witnessed an outstanding progress (*e.g.*, see (Agarwal et al., 2020; Bertsekas, 2000; Jin et al., 2018; Li et al., 2021; Luo et al., 2019; Panait and Luke, 2005; Sidford et al., 2018; Sutton and Barto, 2018), and references therein), the landscape of multi-agent settings eludes a thorough understanding. In fact, guarantees for provably efficient computation of Nash equilibria remain limited to either environments in which agents strive to coordinate towards a shared goal (Chen et al., 2022; Claus and Boutilier, 1998; Ding et al., 2022; Fox et al., 2022; Leonardos et al., 2021; Maheshwari et al., 2022; Wang and Sandholm, 2002; Zhang et al., 2021) or fully competitive such as two-player zero-sum games (Cen et al., 2021; Condon, 1993; Daskalakis et al., 2020; Sayin et al., 2021, 2020; Wei et al., 2021) to name a few. Part of the lack of efficient algorithmic results in MARL is the fact that computing approximate Nash equilibria in (general-sum) games is computationally intractable (Daskalakis et al., 2009; Rubinstein, 2017; Chen et al., 2009; Etessami and Yannakakis, 2010) even when the games have a single state, *i.e.*, normal-form two-player games.

37th Conference on Neural Information Processing Systems (NeurIPS 2023).

We aim at providing a theoretical framework that captures an array of real-world applications with multiple agents — which admittedly correspond to a big portion of all modern applications. A recent contribution that computes NE efficiently in a setting that combines both collaboration and competition, (Kalogiannis et al., 2022), concerns adversarial team Markov games, or competition between an adversary and a group of uncoordinated agents with common rewards. Efficient algorithms for computing Nash equilibria in settings that include both cooperation and competition are far fewer and tend to impose assumptions that are restrictive and difficult to meet in most applications (Bowling, 2000; Hu and Wellman, 2003). The focus of our work is centered around the following question:

> *Are there any other settings of Markov games that encompass both competition and*     $(\star)$
> *coordination while mantaining the tractability of Nash equilibrium computation?*

Inspired by contemporary works in algorithmic game theory and specifically zero-sum *polymatrix* normal-form games (Cai et al., 2016), we focus on the problem of computing Nash equilibria in zero-sum *polymatrix Markov* games. Informally, a polymatrix Markov game is a multi-agent Markov decision process with $n$ agents, state-space $\mathcal{S}$, action space $\mathcal{A}_k$ for agent $k$, a transition probability model $\mathbb{P}$ and is characterized by a graph $\mathcal{G}_s(\mathcal{V}, \mathcal{E}_s)$ which is potentially different in every state $s$. For a fixed state $s$, the nodes of the graph $\mathcal{V}$ correspond to the agents, and the edges $\mathcal{E}_s$ of the graph are two-player normal-form games (different per state). Every node/agent $k$ has a fixed set of actions $\mathcal{A}_k$, and chooses a strategy from this set to play in all games corresponding to adjacent edges. Given an action profile of all the players, the node's reward is the sum of its rewards in all games on the edges adjacent to it. The game is globally zero-sum if, for all strategy profiles, the rewards of all players add up to zero. Afterwards, the process transitions to a state $s'$ according to $\mathbb{P}$. In a more high-level description, the agents interact over a network whose connections change at every state.

**Our results.**     We consider a zero-sum polymatrix Markov game with the additional property that a single agent (not necessarily the same) controls the transition at each state, *i.e.*, the transition model is affected by a single agent's actions for each state $s$. These games are known as *switching controller* Markov games. We show that we can compute in time $\text{poly}(|\mathcal{S}|, n, \max_{i \in [n]} |\mathcal{A}_i|, 1/\epsilon)$ an $\epsilon$-approximate Nash equilibrium. The proof relies on the fact that zero-sum polymatrix Markov games with a switching controller have the following important property: the marginals of a coarse-correlated equilibrium constitute a Nash equilibrium (see Section 3.2). We refer to this phenomenon as *equilibrium collapse*. This property was already known for zero-sum polymatrix normal-form games by Cai et al. (2016) and our results generalize the aforementioned work for Markov games. As a corollary, we get that any algorithm in the literature that guarantees convergence to approximate coarse-correlated equilibria Markovian policies—*e.g.*, (Daskalakis et al., 2022)—can be used to get approximate Nash equilibria. Our contribution also unifies previous results that where otherwise only applicable to the settings of *single* and *switching-control two-player zero-sum games*, or *zero-sum polymatrix normal-form games*. Finally, we show that the equilibrium collapsing phenomenon does not carry over if there are two or more controllers per state (see Section 3.3).

**Technical overview.**     In order to prove our results, we rely on nonlinear programming and, in particular, nonlinear programs whose optima coincide with the Nash equilibria for a particular Markov game (Filar et al., 1991; Filar and Vrieze, 2012). Our approach is analogous to the one used by (Cai et al., 2016) which uses linear programming to prove the collapse of the set of CCE to the set of NE. Nevertheless, using the duality of linear programming in our case is not possible since a Markov game introduces nonlinear terms in the program. It is noteworthy that we do not need to invoke (Lagrangian) duality or an argument that relies on stationary points of a Lagrangian function. Rather, we use the structure of the zero-sum polymatrix Markov games with a switching controller to conclude the relation between a correlated policy and the individual policies formed by its marginals in terms of the individual utilities of the game.

## 1.1 Importance of zero-sum polymatrix Markov games

Strategic interactions of agents over a network is a topic of research in multiple disciplines that span computer science (Easley and Kleinberg, 2010), economics (Schweitzer et al., 2009), control theory (Tipsuwan and Chow, 2003), and biology (Szabó and Fath, 2007) to name a few.

In many environments where multiple agents interact with each other, they do so in a localized manner. That is, every agent is affected by the set of agents that belong to their immediate "neighborhood". Further, it is quite common that these agents will interact independently with each one of their

neighbors; meaning that the outcome of their total interactions is a sum of pairwise interactions rather than interactions that depend on joint actions. Finally, players might remain indifferent to actions of players are not their neighbors.

To illustrate this phenomenon we can think of multiplayer e-games (*e.g.*, CS:GO, Fortnite, League of Legends, etc) where each player interacts through the same move only with players that are present on their premises and, in general, the neighbors cannot combine their actions into something that is not a mere sum of their individual actions (*i.e.,* they rarely can "multiply" the effect of the individual actions). In other scenarios, such as strategic games played on social networks (*e.g.*, opinion dynamics) agents clearly interact in a pairwise manner with agents that belong to their neighborhood and are somewhat oblivious to the actions of agents who they do not share a connection with.

With the proposed model we provide the theoretical framework needed to reason about such strategic interactions over dynamically changing networks.

## 1.2 Related work

From the literature of Markov games, we recognize the settings of *single controller* (Filar and Raghavan, 1984; Sayin et al., 2022; Guan et al., 2016; Qiu et al., 2021) and *switching controller* (Vrieze et al., 1983) Markov games to be one of the most related to ours. In these settings, all agents' actions affect individual rewards, but in every state one particular player (*single controller*), or respectively a potentially different one (*switching controller*), controls the transition of the environment to a new state. To the best of our knowledge, prior to our work, the only Markov games that have been examined under this assumption are either zero-sum or potential games.

Further, we manage to go beyond the dichotomy of absolute competition or absolute collaboration by generalizing zero-sum polymatrix games to their Markovian counterpart. In this sense, our work is related to previous works of Cai et al. (2016); Anagnostides et al. (2022); Ao et al. (2022) which show fast convergence to Nash equilibria in zero-sum polymatrix normal-form games for various no-regret learning algorithms including optimistic gradient descent.

## 2 Preliminaries

**Notation.** We define $[n] \coloneqq \{1, \cdots, n\}$. Scalars are denoted using lightface variables, while, we use boldface for vectors and matrices. For simplicity in the exposition, we use $O(\cdot)$ to suppress dependencies that are polynomial in the parameters of the game. Additionally, given a collection $\boldsymbol{x}$ of policies or strategies for players $[n]$, $\boldsymbol{x}_{-k}$ denotes the policies of every player excluding $k$.

### 2.1 Markov games

In its most general form, a Markov game (MG) with a finite number of $n$ players is defined as a tuple $\Gamma(H, \mathcal{S}, \{\mathcal{A}_k\}_{k\in[n]}, \mathbb{P}, \{r_k\}_{k\in[n]}, \gamma, \boldsymbol{\rho})$. Namely,

- $H \in \mathbb{N}_+$ denotes the *time horizon*, or the length of each episode,
- $\mathcal{S}$, with cardinality $S \coloneqq |\mathcal{S}|$, stands for the state space,
- $\{\mathcal{A}_k\}_{k\in[n]}$ is the collection of every player's action space, while $\mathcal{A} \coloneqq \mathcal{A}_1 \times \cdots \times \mathcal{A}_n$ denotes the *joint action space*; further, an element of that set —a joint action— is generally noted as $\boldsymbol{a} = (a_1, \ldots, a_n) \in \mathcal{A}$,
- $\mathbb{P} \coloneqq \{\mathbb{P}_h\}_{h\in[H]}$ is the set of all *transition matrices*, with $\mathbb{P}_h : \mathcal{S} \times \mathcal{A} \to \Delta(\mathcal{S})$; further, $\mathbb{P}_h(\cdot|s, \boldsymbol{a})$ marks the probability of transitioning to every state given that the joint action $\boldsymbol{a}$ is selected at time $h$ and state $s$ — in infinite-horizon games $\mathbb{P}$ does not depend on $h$ and the index is dropped,
- $r_k \coloneqq \{r_{k,h}\}$ is the reward function of player $k$ at time $h$; $r_{k,h} : \mathcal{S}, \mathcal{A} \to [-1, 1]$ yields the reward of player $k$ at a given state and joint action — in infinite-horizon games, $r_{k,h}$ is the same for every $h$ and the index is dropped,
- a discount factor $\gamma > 0$, which is generally set to 1 when $H < \infty$, and $\gamma < 1$ when $H \to \infty$,
- an initial state distribution $\boldsymbol{\rho} \in \Delta(\mathcal{S})$.

**Policies and value functions.** We will define stationary and nonstationary Markov policies. When the horizon $H$ is finite, a stationary policy equilibrium need not necessarily exist even for a single-agent MG, *i.e.*, a Markov decision process; in this case, we seek nonstationary policies. For the case of infinite-horizon games, it is folklore that a stationary Markov policy Nash equilibrium always exists.

We note that a policy is *Markovian* when it depends on the present state only. A *nonstationary* Markov policy $\boldsymbol{\pi}_k$ for player $k$ is defined as $\boldsymbol{\pi}_k := \{\boldsymbol{\pi}_{k,h} : \mathcal{S} \to \Delta(\mathcal{A}_k), \ \forall h \in [H]\}$. It is a sequence of mappings of states $s$ to a distribution over actions $\Delta(\mathcal{A}_k)$ for every timestep $h$. By $\boldsymbol{\pi}_{k,h}(a|s)$ we will denote the probability of player $k$ taking action $a$ in timestep $h$ and state $s$. A Markov policy is said to be *stationary* in the case that it outputs an identical probability distribution over actions whenever a particular state is visited regardless of the corresponding timestep $h$.

Further, we define a nonstationary Markov *joint policy* $\boldsymbol{\sigma} := \{\boldsymbol{\pi}_h, \ \forall h \in [H]\}$ to be a sequence of mappings from states to distributions over joint actions $\Delta(\mathcal{A}) \equiv \Delta(\mathcal{A}_1 \times \cdots \times \mathcal{A}_n)$ for all times steps $h$ in the time horizon. In this case, the players can be said to share a common source of randomness, or that the joint policy is correlated.

A joint policy $\boldsymbol{\pi}$ will be said to be a *product policy* if there exist policies $\boldsymbol{\pi}_k : [H] \times \mathcal{S} \to \Delta(\mathcal{A}_k), \ \forall k \in [n]$ such that $\boldsymbol{\pi}_h = \boldsymbol{\pi}_{1,h} \times \cdots \times \boldsymbol{\pi}_{n,h}, \ \forall h \in [H]$. Moreover, given a joint policy $\boldsymbol{\pi}$ we let a joint policy $\boldsymbol{\pi}_{-k}$ stand for the *marginal joint policy* excluding player $k$, *i.e.*,

$$\pi_{-k,h}(\boldsymbol{a}|s) = \sum_{a' \in \mathcal{A}_k} \pi_h(a', \boldsymbol{a}|s), \ \forall h \in [H], \forall s \in \mathcal{S}, \forall \boldsymbol{a} \in \mathcal{A}_{-k}.$$

By fixing a joint policy $\boldsymbol{\pi}$ we can define the value function of any given state $s$ and timestep $h$ for every player $k$ as the expected cumulative reward they get from that state and timestep $h$ onward,

$$V_{k,h}^{\boldsymbol{\pi}}(s_1) = \mathbb{E}_{\boldsymbol{\pi}}\left[\sum_{\tau=h}^{H} \gamma^{\tau-1} r_{k,\tau}(s_\tau, \boldsymbol{a}_\tau)\big|s_1\right] = \boldsymbol{e}_{s_1}^\top \sum_{\tau=h}^{H}\left(\gamma^{\tau-1}\prod_{\tau=h}^{h}\mathbb{P}_\tau(\boldsymbol{\pi}_\tau)\right)\boldsymbol{r}_{k,\tau}(\boldsymbol{\pi}_\tau).$$

Depending on whether the game is of finite or infinite horizon we get the followin displays,

- In finite-horizon games, $\gamma = 1$, the value function reads,

$$V_{k,h}^{\boldsymbol{\pi}}(s_1) = \boldsymbol{e}_{s_1}^\top \sum_{\tau=h}^{H}\left(\prod_{\tau'=h}^{\tau}\mathbb{P}_{\tau'}(\boldsymbol{\pi}_{\tau'})\right)\boldsymbol{r}_{k,\tau}(\boldsymbol{\pi}_\tau),$$

- In infinite-horizon games, the value function of each state is,

$$V_k^{\boldsymbol{\pi}}(s_1) = \boldsymbol{e}_{s_1}^\top(\mathbf{I} - \gamma\,\mathbb{P}(\boldsymbol{\pi}))^{-1}\boldsymbol{r}(\boldsymbol{\pi}).$$

Where $\mathbb{P}_h(\boldsymbol{\pi}_h), \mathbb{P}(\boldsymbol{\pi})$ and $\boldsymbol{r}_h(\boldsymbol{\pi}_h), \boldsymbol{r}(\boldsymbol{\pi})$ denote the state-to-state transition probability matrix and expected per-state reward vector for a given policy $\boldsymbol{\pi}_h$ or $\boldsymbol{\pi}$ accordingly. Additionally, $\boldsymbol{e}_{s_1}$ is an all-zero vector apart of a value of 1 in its $s_1$-th position. Also, we denote $V_{k,h}^{\boldsymbol{\pi}}(\boldsymbol{\rho}) = \sum_{s \in \mathcal{S}} \rho(s)V_{k,h}^{\boldsymbol{\pi}}(s)$.

**Best-response policies.** Given an arbitrary joint policy $\boldsymbol{\sigma}$, we define the *best-response policy* of a player $k$ to be a policy $\boldsymbol{\pi}_k^\dagger := \{\boldsymbol{\pi}_{k,h}^\dagger, \ \forall h \in [H]\}$, such that it is a maximizer of $\max_{\boldsymbol{\pi}_k'} V_{k,1}^{\boldsymbol{\pi}_k' \times \boldsymbol{\sigma}_{-k}}(s_1)$. Additionally, we will use the following notation $V_{k,h}^{\dagger,\boldsymbol{\sigma}_{-k}}(s) := \max_{\boldsymbol{\pi}_k'} V_{k,h}^{\boldsymbol{\pi}_k' \times \boldsymbol{\sigma}_{-k}}(s)$.

**Equilibrium notions.** Having defined what a best-response is, it is then quite direct to define different notions of equilibria for Markov games.

**Definition 2.1** (CCE). *We will say that a joint (potentially correlated) policy $\boldsymbol{\sigma} \in \Delta(\mathcal{A})^{H \times S}$ is an $\epsilon$-approximate coarse-correlated equilibrium if it holds that, for an $\epsilon > 0$,*

$$V_{k,1}^{\dagger,\boldsymbol{\sigma}_{-k}}(s_1) - V_{k,1}^{\boldsymbol{\sigma}}(s_1) \leq \epsilon, \ \forall k \in [n]. \tag{CCE}$$

Further, we will define a Nash equilibrium policy,

**Definition 2.2** (NE). *A joint, product policy $\boldsymbol{\pi} \in \prod_{k \in [n]} \Delta(\mathcal{A}_k)^{H \times S}$ is an $\epsilon$-approximate Nash equilibrium if it holds that, for an $\epsilon > 0$,*

$$V_{k,1}^{\dagger,\boldsymbol{\pi}_{-k}}(s_1) - V_{k,1}^{\boldsymbol{\pi}}(s_1) \leq \epsilon, \ \forall k \in [n]. \tag{NE}$$

It is quite evident that an approximate Nash equilibrium is also an approximate coarse-correlated equilibrium while the converse is not generally true. For infinite-horizon games the definitions are analogous and are deferred to the appendix.

## 2.2 Our setting

We focus on the setting of zero-sum polymatrix switching-control Markov games. This setting encompasses two major assumptions related to the reward functions in every state $\{r_k\}_{k\in[n]}$ and the transition kernel $\mathbb{P}$. The first assumption imposes a zero-sum, polymatrix structure on $\{r_k\}_{k\in[n]}$ for every state and directly generalizes zero-sum polymatrix games for games with multiple states.

**Assumption 1** (Zero-sum polymatrix games). *The reward functions of every player in any state $s$ are characterized by a zero-sum, polymatrix structure.*

**Polymatrix structure.**    For every state $s$ there exists an undirected graph $\mathcal{G}_s(\mathcal{V}, \mathcal{E}_s)$ where,

- the set of nodes $\mathcal{V}$ coincides with the set of agents $[n]$; the $k$-th node is the $k$-th agent,
- the set of edges $\mathcal{E}_s$ stands for the set of pair-wise interactions; each edge $e = (k, j), k, j \in [n], k \neq j$ stands for a general-sum normal-form game played between players $k, j$ and which we note as $\left(r_{kj}(s, \cdot, \cdot), r_{jk}(s, \cdot, \cdot)\right)$ with $r_{kj}, r_{jk} : \mathcal{S} \times \mathcal{A}_k \times \mathcal{A}_j \to [-1, 1]$.

Moreover, we define $\mathrm{adj}(s, k) := \{j \in [n] \mid (k, j) \in \mathcal{E}_s\} \subseteq [n]$ to be the set of all neighbors of an arbitrary agent $k$ in state $s$. The reward of agent $k$ at state $s$ given a joint action $\boldsymbol{a}$ depends solely on interactions with their neighbors,

$$r_{k,h}(s, \boldsymbol{a}) = \sum_{j\in\mathrm{adj}(k)} r_{kj,h}(s, a_k, a_j), \; \forall h \in [H], \forall s \in \mathcal{S}, \forall \boldsymbol{a} \in \mathcal{A}.$$

Further, the *zero-sum* assumption implies that,

$$\sum_k r_{k,h}(s, \boldsymbol{a}) = 0, \quad \forall h \in [H], \forall s \in \mathcal{S}, \forall \boldsymbol{a} \in \mathcal{A}. \tag{1}$$

In the infinite-horizon setting, the subscript $h$ can be dropped.

A further assumption (*switching-control*) is necessary in order to ensure the desirable property of equilibrium collapse.

**Assumption 2** (Switching-control). *In every state $s \in \mathcal{S}$, there exists a single player (not necessarily the same), or controller, whose actions determine the probability of transitioning to a new state.*

The function $\mathrm{argctrl} : \mathcal{S} \to [n]$ returns the index of the player who controls the transition probability at a given state $s$. On the other hand, the function $\mathrm{ctrl} : \mathcal{S} \times \mathcal{A} \to \mathcal{A}_{\mathrm{argctrl}(s)}$ gets an input of a joint action $\boldsymbol{a}$, for a particular state $s$, and returns the action of the controller of that state, $a_{\mathrm{argctrl}(s)}$.

**Remark 1.** *It is direct to see that Markov games with a single controller and turn-based Markov games (Daskalakis et al., 2022), are special case of Markov games with switching controller.*

## 3 Main results

In this section we provide the main results of this paper. We shall show the collapsing phenomenon of coarse-correlated equilibria to Nash equilibria in the case of zero-sum, single switching controller polymatrix Markov games. Before we proceed, we provide a formal definition of the notion of collapsing.

**Definition 3.1** (CCE collapse to NE). *Let $\boldsymbol{\sigma}$ be any $\epsilon$-CCE policy of a Markov game. Moreover, let the marginal policy $\boldsymbol{\pi}^{\boldsymbol{\sigma}} := (\boldsymbol{\pi}_1^{\boldsymbol{\sigma}}, ..., \boldsymbol{\pi}_n^{\boldsymbol{\sigma}})$ be defined as:*

$$\pi_k^{\boldsymbol{\sigma}}(a|s) = \sum_{\boldsymbol{a}_{-k}\in\mathcal{A}_{-k}} \sigma(a, \boldsymbol{a}_{-k}|s), \; \forall k, \forall s \in \mathcal{S}, \forall a \in \mathcal{A}_k.$$

*If $\boldsymbol{\pi}^{\boldsymbol{\sigma}}$ is a $O(\epsilon)$-NE equilibrium for every $\boldsymbol{\sigma}$ then we say the set of approximate CCE's collapses to that of approximate NE's.*

We start with the warm-up result that the set of CCE's collapses to the set of NE's for two-player zero-sum Markov games.

## 3.1 Warm-up: equilibrium collapse in two-player zero-sum MG's

Since we focus on two-player zero-sum Markov games, we simplify the notation by using $V_{h=1}^{\cdot}(s) := V_{2,1}^{\cdot}(s)$—*i.e.*, player 1 is the minimizing player and player 2 is the maximizer. We show the following theorem:

**Theorem 3.1** (Collapse in two-player zero-sum MG's). *Let a two-player zero-sum Markov game $\Gamma'$ and an $\epsilon$-approximate CCE policy of that game $\boldsymbol{\sigma}$. Then, the marginalized product policies $\boldsymbol{\pi}_1^{\sigma}, \boldsymbol{\pi}_2^{\sigma}$ form a $2\epsilon$-approximate NE.*

**Proof.** Since $\boldsymbol{\sigma}$ is an $\epsilon$-approximate CCE joint policy, by definition it holds that for any $\boldsymbol{\pi}_1$ and any $\boldsymbol{\pi}_2$,

$$V_{h=1}^{\boldsymbol{\sigma}_{-2} \times \boldsymbol{\pi}_2}(s_1) - \epsilon \leq V_{h=1}^{\boldsymbol{\sigma}}(s_1) \leq V_{h=1}^{\boldsymbol{\pi}_1 \times \boldsymbol{\sigma}_{-1}}(s_1) + \epsilon.$$

Due to Claim A.1, the latter is equivalent to the following inequality,

$$V_{h=1}^{\boldsymbol{\pi}_1^{\sigma} \times \boldsymbol{\pi}_2}(s_1) - \epsilon \leq V_{h=1}^{\boldsymbol{\sigma}}(s_1) \leq V_{h=1}^{\boldsymbol{\pi}_1 \times \boldsymbol{\pi}_2^{\sigma}}(s_1) + \epsilon.$$

Plugging in $\boldsymbol{\pi}_1^{\sigma}, \boldsymbol{\pi}_2^{\sigma}$ alternatingly, we get the inequalities:

$$\begin{cases} V_{h=1}^{\boldsymbol{\pi}_1^{\sigma} \times \boldsymbol{\pi}_2}(s_1) - \epsilon \leq V_{h=1}^{\boldsymbol{\sigma}}(s_1) \leq V_{h=1}^{\boldsymbol{\pi}_1^{\sigma} \times \boldsymbol{\pi}_2^{\sigma}}(s_1) + \epsilon \\ V_{h=1}^{\boldsymbol{\pi}_1^{\sigma} \times \boldsymbol{\pi}_2^{\sigma}}(s_1) - \epsilon \leq V_{h=1}^{\boldsymbol{\sigma}}(s_1) \leq V_{h=1}^{\boldsymbol{\pi}_1 \times \boldsymbol{\pi}_2^{\sigma}}(s_1) + \epsilon \end{cases}$$

The latter leads us to conclude that for any $\boldsymbol{\pi}_1$ and any $\boldsymbol{\pi}_2$,

$$V_{h=1}^{\boldsymbol{\pi}_1^{\sigma} \times \boldsymbol{\pi}_2}(s_1) - 2\epsilon \leq V_{h=1}^{\boldsymbol{\pi}_1^{\sigma} \times \boldsymbol{\pi}_2^{\sigma}}(s_1) \leq V_{h=1}^{\boldsymbol{\pi}_1 \times \boldsymbol{\pi}_2^{\sigma}}(s_1) + 2\epsilon,$$

which is the definition of a NE in a zero-sum game. $\qquad\square$

## 3.2 Equilibrium collapse in finite-horizon polymatrix Markov games

In this section, we focus on the more challenging case of polymatrix Markov games which is the main focus of this paper. For any finite horizon Markov game, we define ($P_{NE}$) to be the following nonlinear program with variables $\boldsymbol{\pi}, \boldsymbol{w}$:

$$\min \sum_{k \in [n]} \left( w_{k,1}(s_1) - \boldsymbol{e}_{s_1}^{\top} \sum_{h=1}^{H} \left( \prod_{\tau=1}^{h} \mathbb{P}_{\tau}(\boldsymbol{\pi}_{\tau}) \right) \boldsymbol{r}_{k,h}(\boldsymbol{\pi}_h) \right)$$

$$\text{s.t. } w_{k,h}(s) \geq r_{k,h}(s, a, \boldsymbol{\pi}_{-k,h}) + \mathbb{P}_h(s, a, \boldsymbol{\pi}_{-k,h}) \boldsymbol{w}_{k,h+1},$$
$$\qquad \forall s \in \mathcal{S}, \forall h \in [H], \forall k \in [n], \forall a \in \mathcal{A}_k;$$
($P_{NE}$)
$$w_{k,H}(s) = 0, \quad \forall k \in [n], \forall s \in \mathcal{S};$$
$$\boldsymbol{\pi}_{k,h}(s) \in \Delta(\mathcal{A}_k),$$
$$\qquad \forall s \in \mathcal{S}, \forall h \in [H], \forall k \in [n], \forall a \in \mathcal{A}_k.$$

Using the following theorem, we are able to use ($P_{NE}$) to argue about equilibrium collapse.

**Theorem 3.2** (NE and global optima of ($P_{NE}$)). *If $(\boldsymbol{\pi}^{\star}, \boldsymbol{w}^{\star})$ yields an $\epsilon$-approximate global minimum of ($P_{NE}$), then $\boldsymbol{\pi}^{\star}$ is an $n\epsilon$-approximate NE of the zero-sum polymatrix switching controller MG, $\Gamma$. Conversely, if $\boldsymbol{\pi}^{\star}$ is an $\epsilon$-approximate NE of the MG $\Gamma$ with corresponding value function vector $\boldsymbol{w}^{\star}$ such that $w_{k,h}^{\star}(s) = V_{k,h}^{\boldsymbol{\pi}^{\star}}(s) \forall (k, h, s) \in [n] \times [H] \times \mathcal{S}$, then $(\boldsymbol{\pi}^{\star}, \boldsymbol{w}^{\star})$ attains an $\epsilon$-approximate global minimum of ($P_{NE}$).*

Following, we are going to use ($P_{NE}$) in proving the collapse of CCE's to NE's. We observe that the latter program is nonlinear and in general nonconvex. Hence, duality cannot be used in the way it was used in (Cai et al., 2016) to prove equilibrium collapse. Nevertheless, we can prove that given a CCE policy $\boldsymbol{\sigma}$, the marginalized, product policy $\bigtimes_{k \in [n]} \boldsymbol{\pi}_k^{\sigma}$ along with an appropriate vector $\boldsymbol{w}^{\sigma}$ achieves a global minimum in the nonlinear program ($P_{NE}$). More precisely, our main result reads as the following statement.

**Theorem 3.3** (CCE collapse to NE in polymatrix MG). *Let a zero-sum polymatrix switching-control Markov game, i.e., a Markov game for which Assumptions 1 and 2 hold. Further, let an $\epsilon$-approximate CCE of that game $\boldsymbol{\sigma}$. Then, the marginal product policy $\boldsymbol{\pi}^{\sigma}$, with $\pi_{k,h}^{\sigma}(a|s) = \sum_{\boldsymbol{a}_{-k} \in \mathcal{A}_{-k}} \boldsymbol{\sigma}_h(a, \boldsymbol{a}_{-k})$, $\forall k \in [n], \forall h \in [H]$ is an $n\epsilon$-approximate NE.*

**Proof.** Let an $\epsilon$-approximate CCE policy, $\boldsymbol{\sigma}$, of game $\Gamma$. Moreover, let the best-response value-vectors of each agent $k$ to joint policy $\boldsymbol{\sigma}_{-k}$, $\boldsymbol{w}_k^{\dagger}$.

Now, we observe that due to Assumption 1,

$$
\begin{aligned}
w_{k,h}^{\dagger}(s) &\geq r_{k,h}(s, a, \boldsymbol{\sigma}_{-k,h}) + \mathbb{P}_h(s, a, \boldsymbol{\sigma}_{-k,h})\boldsymbol{w}_{k,h+1}^{\dagger} \\
&= \sum_{j \in \mathrm{adj}(k)} r_{(k,j),h}(s, a, \boldsymbol{\pi}_j^{\sigma}) + \mathbb{P}_h(s, a, \boldsymbol{\sigma}_{-k,h})\boldsymbol{w}_{k,h+1}^{\dagger}.
\end{aligned}
$$

Further, due to Assumption 2,

$$
\mathbb{P}_h(s, a, \boldsymbol{\sigma}_{-k,h})\boldsymbol{w}_{k,h+1}^{\dagger} = \mathbb{P}_h(s, a, \boldsymbol{\pi}_{\mathrm{argctrl}(s),h}^{\sigma})\boldsymbol{w}_{k,h+1}^{\dagger},
$$

or,

$$
\mathbb{P}_h(s, a, \boldsymbol{\sigma}_{-k,h})\boldsymbol{w}_{k,h+1}^{\dagger} = \mathbb{P}_h(s, a, \boldsymbol{\pi}^{\sigma})\boldsymbol{w}_{k,h+1}^{\dagger}.
$$

Putting these pieces together, we reach the conclusion that $(\boldsymbol{\pi}^{\sigma}, \boldsymbol{w}^{\dagger})$ is feasible for the nonlinear program ($\mathrm{P_{NE}}$).

What is left is to prove that it is also an $\epsilon$-approximate global minimum. Indeed, if $\sum_k \boldsymbol{w}_{k,h}^{\dagger}(s_1) \leq \epsilon$ (by assumption of an $\epsilon$-approximate CCE), then the objective function of ($\mathrm{P_{NE}}$) will attain an $\epsilon$-approximate global minimum. In turn, due to Theorem 3.2 the latter implies that $\boldsymbol{\pi}^{\sigma}$ is an $n\epsilon$-approximate NE. $\qquad\square$

We can now conclude that due to the algorithm introduced in (Daskalakis et al., 2022) for CCE computation in general-sum MG's, the next statement holds true.

**Corollary 3.1** (Computing a NE—finite-horizon). Given a finite-horizon switching control zero-sum polymatrix Markov game, we can compute an $\epsilon$-approximate Nash equilibrium policy that is Markovian with probability at least $1 - \delta$ in time $\mathrm{poly}\left(n, H, S, \max_k |\mathcal{A}_k|, \frac{1}{\epsilon}, \log(1/\delta)\right)$.

In the next section, we discuss the necessity of the assumption of switching control using a counter-example of non-collapsing equilibria.

## 3.3 No equilibrium collapse with more than one controllers per-state

Although Assumption 1 is sufficient for the collapse of any CCE to a NE in single-state (*i.e.*, normal-form) games, we will prove that Assumption 2 is indispensable in guaranteeing such a collapse in zero-sum polymatrix Markov games. That is, if more than one players affect the transition probability from one state to another, a CCE is not guaranteed to collapse to a NE.

**Example 1.** *We consider the following 3-player Markov game that takes place for a time horizon $H = 3$. There exist three states, $s_1, s_2,$ and $s_3$ and the game starts at state $s_1$. Player 3 has a single action in every state, while players 1 and 2 have two available actions $\{a_1, a_2\}$ and $\{b_1, b_2\}$ respectively in every state.*

**Reward functions.** *If player 1 (respectively, player 2) takes action $a_1$ (resp., $b_1$), in either of the states $s_1$ or $s_2$, they get a reward equal to $\frac{1}{20}$. In state $s_3$, both players get a reward equal to $-\frac{1}{2}$ regardless of the action they select. Player 3 always gets a reward that is equal to the negative sum of the reward of the other two players. This way, the* zero-sum polymatrix property *of the game is ensured (Assumption 1).*

**Transition probabilities.** *If players $1$ and $2$ select the joint action $(a_1, b_1)$ in state $s_1$, the game will transition to state $s_2$. In any other case, it will transition to state $s_3$. The converse happens if in state $s_2$ they take joint action $(a_1, b_1)$; the game will transition to state $s_3$. For any other joint action, it will transition to state $s_1$. From state $s_3$, the game transitions to state $s_1$ or $s_2$ uniformly at random.*

*At this point, it is important to notice that two players control the transition probability from one state to another. In other words, Assumption 2 does not hold.*

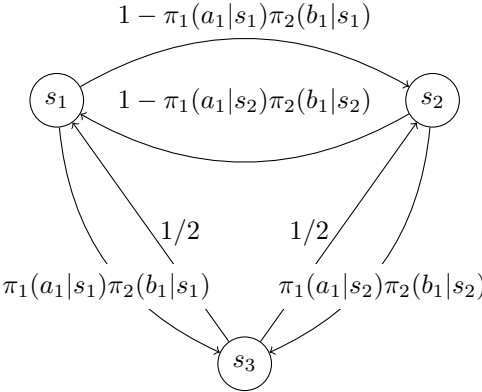

Figure 1: A graph of the state space with transition probabilities parametrized with respect to the policy of each player.

*Next, we consider the joint policy $\boldsymbol{\sigma}$,*

$$\boldsymbol{\sigma}(s_1) = \boldsymbol{\sigma}(s_2) = \begin{array}{c} \\ a_1 \\ a_2 \end{array} \begin{array}{c} b_1 \qquad b_2 \\ \begin{pmatrix} 0 & 1/2 \\ 1/2 & 0 \end{pmatrix}. \end{array}$$

**Claim 3.1.** The joint policy $\boldsymbol{\sigma}$ that assigns probability $\frac{1}{2}$ to the joint actions $(a_1, b_2)$ and $(a_2, b_1)$ in both states $s_1, s_2$ is a CCE and $V_{1,1}^{\boldsymbol{\sigma}}(s_1) = V_{2,1}^{\boldsymbol{\sigma}}(s_1) = \frac{1}{20}$.

*Yet, the marginalized product policy of $\boldsymbol{\sigma}$ which we note as $\boldsymbol{\pi}_1^{\boldsymbol{\sigma}} \times \boldsymbol{\pi}_2^{\boldsymbol{\sigma}}$ does not constitute a NE. The components of this policy are,*

$$\begin{cases} \boldsymbol{\pi}_1^{\boldsymbol{\sigma}}(s_1) = \boldsymbol{\pi}_1^{\boldsymbol{\sigma}}(s_2) = \begin{array}{c} a_1 \qquad a_2 \\ \begin{pmatrix} 1/2 & 1/2 \end{pmatrix} \end{array}, \\[2em] \boldsymbol{\pi}_2^{\boldsymbol{\sigma}}(s_1) = \boldsymbol{\pi}_2^{\boldsymbol{\sigma}}(s_2) = \begin{array}{c} b_1 \qquad b_2 \\ \begin{pmatrix} 1/2 & 1/2 \end{pmatrix} \end{array}. \end{cases}$$

*I.e., the product policy $\boldsymbol{\pi}_1^{\boldsymbol{\sigma}} \times \boldsymbol{\pi}_2^{\boldsymbol{\sigma}}$ selects any of the two actions of each player in states $s_1, s_2$ independently and uniformally at random. With the following claim, it can be concluded that in general when more than one player control the transition the set of equilibria do not collapse.*

**Claim 3.2.** The product policy $\boldsymbol{\pi}_1^{\boldsymbol{\sigma}} \times \boldsymbol{\pi}_2^{\boldsymbol{\sigma}}$ is not a NE.

*In conclusion, Assumption 1 does not suffice to ensure equilibrium collapse.*

**Theorem 3.4.** *There exists a zero-sum polymatrix Markov game (Assumption 2 is not satisfied) that has a CCE which does not collapse to a NE.*

### 3.4 Equilibrium collapse in infinite-horizon polymatrix Markov games

In proving equilibrium collapse for infinite-horizon polymatrix Markov games, we use similar arguments and the following nonlinear program with variables $\boldsymbol{\pi}, \boldsymbol{w}$,

$$\min \sum_{k \in [n]} \boldsymbol{\rho}^\top \left( \boldsymbol{w}_k - (\mathbf{I} - \gamma \, \mathbb{P}(\boldsymbol{\pi}))^{-1} \boldsymbol{r}_k(\boldsymbol{\pi}) \right)$$

$$\text{s.t. } w_k(s) \geq r_k(s, a, \boldsymbol{\pi}_{-k}) + \gamma \, \mathbb{P}(s, a, \boldsymbol{\pi}_{-k}) \boldsymbol{w}_k,$$

(P$'_{\text{NE}}$)
$$\forall s \in \mathcal{S}, \forall k \in [n], \forall a \in \mathcal{A}_k;$$
$$\boldsymbol{\pi}_k(s) \in \Delta(\mathcal{A}_k),$$
$$\forall s \in \mathcal{S}, \forall k \in [n], \forall a \in \mathcal{A}_k.$$

We note that Example 1 can be properly adjusted to show that the switching-control assumption is necessary for equilibrium collapse in infinite-horizon games as well. Compared to finite-horizon games, infinite-horizon games cannot be possibly solved using backward induction. They pose a genuine computational challenge and, in that sense, the importance of the property of equilibrium collapse gets highlighted.

**Computational implications.** Equilibrium collapse in infinite-horizon MG's allows us to use the CCE computation technique found in (Daskalakis et al., 2022) in order to compute an $\epsilon$-approximate NE. Namely, given an accuracy threshold $\epsilon$, we truncate the infinite-horizon game to its *effective horizon* $H := \frac{\log(1/\epsilon)}{1-\gamma}$. Then, we define reward functions that depend on the time-step $h$, *i.e.*, $r_{k,h} = \gamma^{h-1} r_k$. Finally,

**Corollary 3.2.** (Computing a NE—infinite-horizon) Given an infinite-horizon switching control zero-sum polymatrix game $\Gamma$, it is possible to compute a Nash equilibrium policy that is Markovian and nonstationary with probability at least $1 - \delta$ in time $\text{poly}\left( n, \frac{1}{1-\gamma}, S, \max_k |\mathcal{A}_k|, \frac{1}{\epsilon}, \log(1/\delta) \right)$.

## 4 Conclusion and open problems

In this paper, we unified switching-control Markov games and zero-sum polymatrix normal-form games. We highlighted how numerous applications can be modeled using this framework and we focused on the phenomenon of equilibrium collapse from the set of coarse-correlated equilibria to that of Nash equilibria. This property holds implications for computing approximate Nash equilibria in switching control zero-sum polymatrix Markov games; it ensures that it can be done efficiently.

**Open problems.** In light of the proposed problem and our results there are multiple interesting open questions:

- Is it possible to use a policy optimization algorithm similar to those of (Erez et al., 2022; Zhang et al., 2022) in order to converge to an approximate Nash equilibrium? We note that the question can be settled in one of two ways; *either* extend the current result of equilibrium collapse to policies that are non-Markovian *or* guarantee convergence to Markovian policies. The notion of *regret* in (Erez et al., 2022) gives rise to the computation of a CCE that is a non-Markovian policy in the sense that the policy at every timestep depends on the policy sampled from the history of no-regret play and not only the given state.

- We conjecture that a convergence rate of $O(\frac{1}{T})$ to a NE is possible, *i.e.*, there exists an algorithm with running time $O(1/\epsilon)$ that computes an $\epsilon$-approximate NE.

- Are there more classes of Markov games in which computing Nash equilibria is computationally tractable?

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

# A  Missing statements and proofs

## A.1  Statements for Section 3.1

**Claim A.1.** Let a two-player Markov game where both players affect the transition. Further, consider a correlated policy $\boldsymbol{\sigma}$ and its corresponding marginalized product policy $\boldsymbol{\pi}^{\boldsymbol{\sigma}} = \boldsymbol{\pi}_1^{\boldsymbol{\sigma}} \times \boldsymbol{\pi}_2^{\boldsymbol{\sigma}}$. Then, for any $\boldsymbol{\pi}_1', \boldsymbol{\pi}_2'$,

$$V_{k,1}^{\boldsymbol{\pi}_1', \boldsymbol{\sigma}_{-1}}(s_1) = V_{k,1}^{\boldsymbol{\pi}_1', \boldsymbol{\pi}_2^{\boldsymbol{\sigma}}}(s_1),$$

$$V_{k,2}^{\boldsymbol{\sigma}_{-2}, \boldsymbol{\pi}_2'}(s_1) = V_{k,2}^{\boldsymbol{\pi}_1^{\boldsymbol{\sigma}}, \boldsymbol{\pi}_2'}(s_1).$$

**Proof.** We will effectively show that the problem of best-responding to a correlated policy $\boldsymbol{\sigma}$ is equivalent to best-responding to the marginal policy of $\boldsymbol{\sigma}$ for the opponent. The proof follows from the equivalence of the two MDPs.

As a reminder,

$$\pi_{1,h}(a|s) = \sum_{b \in \mathcal{A}_2} \boldsymbol{\sigma}_h(a, b|s)$$

$$\pi_{2,h}(b|s) = \sum_{a \in \mathcal{A}_1} \boldsymbol{\sigma}_h(a, b|s)$$

As we have seen in Section 2.1, in the case of unilateral deviation from joint policy $\boldsymbol{\sigma}$, an agent faces a single agent MDP. More specifically, agent 2, best-responds by optimizing a reward function $\bar{r}_{2,h}(s, b)$ under a transition kernel $\bar{\mathbb{P}}_2$ for which,

$$\bar{r}_{2,h}(s, b) = \mathbb{E}_{b \sim \boldsymbol{\sigma}}\left[r_{2,h}(s, a, b)\right] = \mathbb{E}_{b \sim \boldsymbol{\pi}_1^{\boldsymbol{\sigma}}}\left[r_{2,h}(s, a, b)\right] = r_{2,h}(s, \boldsymbol{\pi}_1^{\boldsymbol{\sigma}}, b).$$

Similarly,

$$\bar{r}_{1,h}(s, b) = r_{1,h}(s, a, \boldsymbol{\pi}_2^{\boldsymbol{\sigma}}).$$

Analogously, for each of the transition kernels,

$$\bar{\mathbb{P}}_{2,h}(s'|s, b) = \mathbb{E}_{a \sim \boldsymbol{\sigma}}\left[\mathbb{P}_{2,h}(s'|s, a, b)\right] = \mathbb{E}_{a \sim \boldsymbol{\pi}_2^{\boldsymbol{\sigma}}}\left[\mathbb{P}_{2,h}(s'|s, a, b)\right] = \mathbb{P}_{2,h}(s'|s, \boldsymbol{\pi}_1^{\boldsymbol{\sigma}}, b),$$

as for agent 1,

$$\bar{\mathbb{P}}_{1,h}(s'|s, a) = \mathbb{P}_{1,h}(s'|s, a, \boldsymbol{\pi}_2^{\boldsymbol{\sigma}}).$$

Hence, it follows that, $V_{2,1}^{\boldsymbol{\sigma}_{-2} \times \boldsymbol{\pi}_2'}(s_1) = V_{2,1}^{\boldsymbol{\pi}_1^{\boldsymbol{\sigma}} \times \boldsymbol{\pi}_2'}(s_1)$, $\forall \boldsymbol{\pi}_2'$ and $V_{1,1}^{\boldsymbol{\pi}_1' \times \boldsymbol{\sigma}_{-1}}(s_1) = V_{1,1}^{\boldsymbol{\pi}_1' \times \boldsymbol{\pi}_2^{\boldsymbol{\sigma}}}(s_1)$, $\forall \boldsymbol{\pi}_2'$.

$\square$

## A.2  Proof of Theorem 3.2

**The best-response program.** First, we state the following lemma that will prove useful for several of our arguments,

**Lemma A.1** (Best-response LP). Let a (possibly correlated) joint policy $\hat{\boldsymbol{\sigma}}$. Consider the following linear program with variables $\boldsymbol{w} \in \mathbb{R}^{n \times H \times S}$,

$$\begin{aligned}
\min \quad & \sum_{k \in [n]} w_{k,s}(s_1) - \boldsymbol{e}_{s_1}^{\top} \sum_{h=1}^{H} \left(\prod_{\tau=1}^{h} \mathbb{P}_{\tau}(\hat{\boldsymbol{\sigma}}_{\tau})\right) \boldsymbol{r}_{k,h}(\hat{\boldsymbol{\sigma}}_h) \\
\text{s.t.} \quad & w_{k,h}(s) \geq r_{k,h}(s, a, \hat{\boldsymbol{\sigma}}_{-k,h}) + \mathbb{P}_h(s, a, \hat{\boldsymbol{\sigma}}_{-k,h}) \boldsymbol{w}_{k,h+1}, \\
& \qquad \forall s \in \mathcal{S}, \forall h \in [H], \forall k \in [n], \forall a \in \mathcal{A}_k; \\
& w_{k,H}(s) = 0, \ \forall k \in [n], \forall s \in \mathcal{S}.
\end{aligned}$$

(P$_{\text{BR}}$)

The optimal solution $\boldsymbol{w}^{\dagger}$ of the program is unique and corresponds to the value function of each player $k \in [n]$ when player $k$ best-responds to $\hat{\boldsymbol{\sigma}}$.

**Proof.** We observe that the program is separable to $n$ independent linear programs, each with variables $\boldsymbol{w}_k \in \mathbb{R}^{n \times H}$,

$$\min \ w_{k,1}(s_1)$$
$$\text{s.t. } w_{k,h}(s) \geq r_{k,h}(s, a, \hat{\boldsymbol{\sigma}}_{-k,h}) + \mathbb{P}_h(s, a, \hat{\boldsymbol{\sigma}}_{-k,h}) \boldsymbol{w}_{k,h+1},$$
$$\forall s \in \mathcal{S}, \forall h \in [H], \forall a \in \mathcal{A}_k;$$
$$w_{k,H}(s) = 0, \ \forall k \in [n], \forall s \in \mathcal{S}.$$

Each of these linear programs describes the problem of a single agent MDP (Neu and Pike-Burke, 2020, Section 2) —that agent being $k$— which, as we have seen in Best-response policies, is equivalent to the problem of finding a best-response to $\hat{\boldsymbol{\sigma}}_{-k}$. It follows that the optimal $\boldsymbol{w}_k^\dagger$ for every program is unique (each program corresponds to a set of Bellman optimality equations). $\qquad\square$

**Properties of the NE program.** Second, we need to prove that the minimum value of the objective function of the program is nonnegative.

**Lemma A.2** (Feasibility of ($P'_{\text{NE}}$) and global optimum)**.** The nonlinear program ($P'_{\text{NE}}$) is feasible, has a nonnegative objective value, and its global minimum is equal to $0$.

**Proof.** Analogously to the finite-horizon case, for the feasibility of the nonlinear program, we invoke the theorem of the existence of a Nash equilibrium. We let a NE product policy, $\boldsymbol{\pi}^\star$, and a vector $\boldsymbol{w}^\star \in \mathbb{R}^{n \times S}$ such that $w_k^\star(s) = V_k^{\dagger, \boldsymbol{\pi}^\star_{-k}}(s), \ \forall k \in [n] \times \mathcal{S}$.

By Lemma A.1, we know that $(\boldsymbol{\pi}^\star, \boldsymbol{w}^\star)$ satisfies all the constraints of ($P_{\text{NE}}$). Additionaly, because $\boldsymbol{\pi}^\star$ is a NE, $V_{k,h}^{\boldsymbol{\pi}^\star}(s_1) = V_{k,h}^{\dagger, \boldsymbol{\pi}^\star_{-k}}(s_1)$ for all $k \in [n]$. Observing that,

$$w_{k,1}^\star(s_1) - \boldsymbol{e}_{s_1}^\top \sum_{h=1}^H \left( \prod_{\tau=1}^h \mathbb{P}_\tau(\boldsymbol{\pi}_\tau^\star) \right) \boldsymbol{r}_{k,h}(\boldsymbol{\pi}_h^\star) = V_{k,h}^{\dagger, \boldsymbol{\pi}^\star_{-k}}(s_1) - V_{k,h}^{\boldsymbol{\pi}^\star}(s_1) = 0,$$

concludes the argument that a NE attains an objective value equal to $0$.

Continuing, we observe that due to (1) the objective function can be equivalently rewritten as,

$$\sum_{k \in [n]} \left( w_{k,1}(s_1) - \boldsymbol{e}_{s_1}^\top \sum_{h=1}^H \left( \prod_{\tau=1}^h \mathbb{P}_\tau(\boldsymbol{\pi}_\tau) \right) \boldsymbol{r}_{k,h}(\boldsymbol{\pi}_h) \right)$$
$$= \sum_{k \in [n]} w_{k,1}(s_1) - \boldsymbol{e}_{s_1}^\top \sum_{h=1}^H \left( \prod_{\tau=1}^h \mathbb{P}_\tau(\boldsymbol{\pi}_\tau) \right) \sum_{k \in [n]} \boldsymbol{r}_{k,h}(\boldsymbol{\pi}_h)$$
$$= \sum_{k \in [n]} w_{k,1}(s_1).$$

Next, we focus on the inequality constraint

$$w_{k,h}(s) \geq r_{k,h}(s, a, \boldsymbol{\pi}_{-k,h}) + \mathbb{P}_h(s, a, \boldsymbol{\pi}_{-k,h}) \boldsymbol{w}_{k,h+1}$$

which holds for all $s \in \mathcal{S}$, all players $k \in [n]$, all $a \in \mathcal{A}_k$, and all timesteps $h \in [H-1]$.

By summing over $a \in \mathcal{A}_k$ while multiplying each term with a corresponding coefficient $\pi_{k,h}(a|s)$, the display written in an equivalent element-wise vector inequality reads:

$$\boldsymbol{w}_{k,h} \geq \boldsymbol{r}_{k,h}(\boldsymbol{\pi}_h) + \mathbb{P}_h(\boldsymbol{\pi}_h) \boldsymbol{w}_{k,h+1}.$$

Finally, after consecutively substituting $\boldsymbol{w}_{k,h+1}$ with the element-wise lesser term $\boldsymbol{r}_{k,h+1}(\boldsymbol{\pi}_{h+1}) + \mathbb{P}_{h+1}(\boldsymbol{\pi}_{h+1}) \boldsymbol{w}_{k,h+2}$, we end up with the inequality:

$$\boldsymbol{w}_{k,1} \geq \sum_{h=1}^H \left( \prod_{\tau=1}^h \mathbb{P}_\tau(\boldsymbol{\pi}_\tau) \right) \boldsymbol{r}_{k,h}(\boldsymbol{\pi}_h). \tag{5}$$

Summing over $k$, it holds for the $s_1$-th entry of the inequality,

$$\sum_{k \in [n]} w_{k,1} \geq \sum_{k \in [n]} \sum_{h=1}^H \left( \prod_{\tau=1}^h \mathbb{P}_\tau(\boldsymbol{\pi}_\tau) \right) \boldsymbol{r}_{k,h}(\boldsymbol{\pi}_h) = 0.$$

Where the equality holds due to the zero-sum property, (1). $\qquad\square$

**An approximate NE is an approximate global minimum.** We show that an $\epsilon$-approximate NE, $\boldsymbol{\pi}^\star$, achieves an $n\epsilon$-approximate global minimum of the program. Utilizing Lemma A.1, setting $w_k^\star(s_1) = V_{k,1}^{\dagger, \boldsymbol{\pi}^\star_{-k}}(s_1)$, and the definition of an $\epsilon$-approximate NE we see that,

$$\sum_{k \in [n]} \left( w_{k,1}^\star(s_1) - \boldsymbol{e}_{s_1}^\top \sum_{h=1}^H \left( \prod_{\tau=1}^h \mathbb{P}_\tau(\boldsymbol{\pi}_\tau^\star) \right) \boldsymbol{r}_{k,h}(\boldsymbol{\pi}_h^\star) \right) = \sum_{k \in [n]} \left( w_{k,1}^\star(s_1) - V_{k,1}^{\boldsymbol{\pi}^\star}(s_1) \right)$$

$$\leq \sum_{k \in [n]} \epsilon = n\epsilon.$$

Indeed, this means that $\boldsymbol{\pi}^\star, \boldsymbol{w}^\star$ is an $n\epsilon$-approximate global minimizer of ($\mathrm{P}_{\mathrm{NE}}$).

**An approximate global minimum is an approximate NE.** For the opposite direction, we let a feasible $\epsilon$-approximate global minimizer of the program ($\mathrm{P}_{\mathrm{NE}}$), $(\boldsymbol{\pi}^\star, \boldsymbol{w}^\star)$. Because a global minimum of the program is equal to 0, an $\epsilon$-approximate global optimum must be at most $\epsilon > 0$. We observe that for every $k \in [n]$,

$$w_{k,1}^\star(s_1) \geq \boldsymbol{e}_{s_1}^\top \sum_{h=1}^H \left( \prod_{\tau=1}^h \mathbb{P}_\tau(\boldsymbol{\pi}_\tau^\star) \right) \boldsymbol{r}_{k,h}(\boldsymbol{\pi}_h^\star), \tag{6}$$

which follows from induction on the inequality constraint over all $h$ similar to (5).

Consequently, the assumption that

$$\epsilon \geq \sum_{k \in [n]} \left( w_{k,1}^\star(s_1) - \boldsymbol{e}_{s_1}^\top \sum_{h=1}^H \left( \prod_{\tau=1}^h \mathbb{P}_\tau(\boldsymbol{\pi}_\tau^\star) \right) \boldsymbol{r}_{k,h}(\boldsymbol{\pi}_h^\star) \right),$$

and Equation (6), yields the fact that

$$\epsilon \geq w_{k,1}^\star(s_1) - \boldsymbol{e}_{s_1}^\top \sum_{h=1}^H \left( \prod_{\tau=1}^h \mathbb{P}_\tau(\boldsymbol{\pi}_\tau^\star) \right) \boldsymbol{r}_{k,h}(\boldsymbol{\pi}_h^\star)$$

$$\geq V_{k,1}^{\dagger, \boldsymbol{\pi}^\star_{-k}}(s_1) - V_{k,1}^{\boldsymbol{\pi}^\star}(s_1),$$

where the second inequality holds from the fact that $\boldsymbol{w}^\star$ is feasible for ($\mathrm{P}_{\mathrm{BR}}$). The latter concludes the proof, as the display coincides with the definition of an $\epsilon$-approximate NE.

### A.3 Proof of Claim 3.1

**Proof.** The value function of $s_1$ for $h = 1$ of players 1 and 2 read:

$$V_{1,1}^{\boldsymbol{\sigma}}(s_1) = \boldsymbol{e}_{s_1}^\top \left( \boldsymbol{r}_1(\boldsymbol{\sigma}) + \mathbb{P}(\boldsymbol{\sigma})\boldsymbol{r}_1(\boldsymbol{\sigma}) \right)$$

$$= -\frac{9\sigma(a_1, b_1|s_1)}{20} + \frac{\sigma(a_1, b_2|s_1)}{20} + \frac{(1 - \sigma(a_1, b_1|s_1))(\sigma(a_1, b_1|s_2) + \sigma(a_1, b_2|s_2))}{20},$$

and,

$$V_{2,1}^{\boldsymbol{\sigma}}(s_1) = \boldsymbol{e}_{s_1}^\top \left( \boldsymbol{r}_2(\boldsymbol{\sigma}) + \mathbb{P}(\boldsymbol{\sigma})\boldsymbol{r}_2(\boldsymbol{\sigma}) \right)$$

$$= -\frac{9\sigma(a_1, b_1|s_1)}{20} + \frac{\sigma(a_2, b_2|s_1)}{20} + \frac{(1 - \sigma(a_1, b_1|s_1))(\sigma(a_1, b_1|s_2) + \sigma(a_2, b_1|s_2))}{20}.$$

We are indifferent to the corresponding value function of player 3 as they only have one available action per state and hence, cannot affect their rewards. For the joint policy $\boldsymbol{\sigma}$, the corresponding value functions of both players 1 and 2 are $V_{1,1}^{\boldsymbol{\sigma}}(s_1) = V_{2,1}^{\boldsymbol{\sigma}}(s_1) = \frac{1}{20}$.

**Deviations.** We will now prove that no deviation of player 1 manages to accumulate a reward greater than $\frac{1}{20}$. The same follows for player 2 due to symmetry.

When a player deviates unilaterally from a joint policy, they experience a single agent Markov decision process (MDP). It is well-known that MDPs always have a deterministic optimal policy. As such, it suffices to check whether $V_{1,1}^{\boldsymbol{\pi}_1, \boldsymbol{\sigma}_{-1}}(s_1)$ is greater than $\frac{1}{20}$ for any of the four possible deterministic policies:

- $\boldsymbol{\pi}_1(s_1) = \boldsymbol{\pi}_1(s_2) = (1 \quad 0)$,
- $\boldsymbol{\pi}_1(s_1) = \boldsymbol{\pi}_1(s_2) = (0 \quad 1)$,

- $\boldsymbol{\pi}_1(s_1) = (1 \quad 0)$, $\boldsymbol{\pi}_1(s_2) = (0 \quad 1)$,
- $\boldsymbol{\pi}_1(s_1) = (0 \quad 1)$, $\boldsymbol{\pi}_1(s_2) = (1 \quad 0)$.

Finally, the value function of any deviation $\boldsymbol{\pi}_1'$ writes,

$$V_{1,1}^{\boldsymbol{\pi}_1' \times \boldsymbol{\sigma}_{-1}}(s_1) = -\frac{\pi_1'(a_1|s_1)}{5} - \frac{\pi_1'(a_1|s_2)\,(\pi_1'(a_1|s_1) - 2)}{40}.$$

We can now check that for all deterministic policies $V_{1,1}^{\boldsymbol{\pi}_1' \times \boldsymbol{\sigma}_{-1}}(s_1) \leq \frac{1}{20}$. By symmetry, it follows that $V_{2,1}^{\boldsymbol{\pi}_2' \times \boldsymbol{\sigma}_{-2}}(s_1) \leq \frac{1}{20}$ and as such $\boldsymbol{\sigma}$ is indeed a CCE. $\qquad\square$

### A.4 Proof of Claim 3.2

**Proof.** In general, the value functions of each player 1 and 2 are:

$$V_{1,1}^{\boldsymbol{\pi}_1 \times \boldsymbol{\pi}_2}(s_1) = -\frac{\pi_1(a_1|s_1)\pi_2(b_1|s_1)}{2} + \frac{\pi_1(a_1|s_1)}{20} - \frac{\pi_1(a_1|s_2)\,(\pi_1(a_1|s_1)\pi_2(b_1|s_1) - 1)}{20},$$

and

$$V_{2,1}^{\boldsymbol{\pi}_1 \times \boldsymbol{\pi}_2}(s_1) = -\frac{\pi_1(a_1|s_1)\pi_2(b_1|s_1)}{2} + \frac{\pi_1(b_1|s_1)}{20} - \frac{\pi_1(b_1|s_2)\,(\pi_1(a_1|s_1)\pi_2(b_1|s_1) - 1)}{20}.$$

Plugging in $\boldsymbol{\pi}_1^\sigma, \boldsymbol{\pi}_2^\sigma$ yields $V_{1,1}^{\boldsymbol{\pi}_1^\sigma \times \boldsymbol{\pi}_2^\sigma}(s_1) = V_{2,1}^{\boldsymbol{\pi}_1^\sigma \times \boldsymbol{\pi}_2^\sigma}(s_1) = -\frac{13}{160}$. But, if player 1 deviates to say $\pi_1'(s_1) = \pi_1'(s_2) = (0 \quad 1)$, they get a value equal to 0 which is clearly greater than $-\frac{13}{160}$. Hence, $\boldsymbol{\pi}_1^\sigma \times \boldsymbol{\pi}_2^\sigma$ is not a NE. $\qquad\square$

### A.5 Proof of Theorem 3.4

**Proof.** The proof follows from the game of Example 1, and Claims 3.1 and 3.2. $\qquad\square$

# B    Proofs for infinite-horizon Zero-Sum Polymatrix Markov Games

In this section we will explicitly state definitions, theorems and proofs relating to the infinite-horizon discounted zero-sum polymatrix Markov games.

## B.1    Definitions of equilibria for the infinite-horizon

Let us restate the definition specifically for infinite-horizon Markov games. They are defined as a tuple $\Gamma(H, \mathcal{S}, \{\mathcal{A}_k\}_{k \in [n]}, \mathbb{P}, \{r_k\}_{k \in [n]}, \gamma, \boldsymbol{\rho})$.

- $H = \infty$ denotes the *time horizon*
- $\mathcal{S}$, with cardinality $S := |\mathcal{S}|$, stands for the state space,
- $\{\mathcal{A}_k\}_{k \in [n]}$ is the collection of every player's action space, while $\mathcal{A} := \mathcal{A}_1 \times \cdots \times \mathcal{A}_n$ denotes the *joint action space*; further, an element of that set —a joint action— is generally noted as $\boldsymbol{a} = (a_1, \ldots, a_n) \in \mathcal{A}$,
- $\mathbb{P} : \mathcal{S} \times \mathcal{A} \to \Delta(\mathcal{S})$ is the transition probability function,
- $r_k : \mathcal{S}, \mathcal{A} \to [-1, 1]$ yields the reward of player $k$ at a given state and joint action,
- a discount factor $0 < \gamma < 1$,
- an initial state distribution $\boldsymbol{\rho} \in \Delta(\mathcal{S})$.

**Policies and value functions.**    In infinite-horizon Markov games policies can still be distinguished in two main ways, *Markovian/non-Markovian* and *stationary/nonstationary*. Moreover, a joint policy can be a *correlated* policy or a *product* policy.

*Markovian* policies attribute a probability over the simplex of actions solely depending on the running state $s$ of the game. On the other hand, *non-Markovian* policies attribute a probability over the simplex of actions that depends on any subset of the history of the game. *I.e.*, they can depend on any sub-sequence of actions and states up until the running timestep of the horizon.

*Stationary* policies are those that will attribute the same probability distribution over the simplex of actions for every timestep of the horizon. *Nonstationary* policies, on the contrary can change depending on the timestep of the horizon.

A joint Markovian stationary policy $\boldsymbol{\sigma}$ is said to be *correlated* when for every state $s \in \mathcal{S}$, attributes a probability distribution over the simplex of joint actions $\mathcal{A}$ for all players, *i.e.*, $\boldsymbol{\sigma}(s) \in \Delta(\mathcal{A})$. A Markovian stationary policy $\boldsymbol{\pi}$ is said to be a *product* policy when for every $s \in \mathcal{S}$, $\boldsymbol{\pi}(s) \in \prod_{k=1}^n \Delta(\mathcal{A}_k)$. It is rather easy to define *correlated/product* policies for the case of non-Markovian and nonstationary policies.

Given a Markovian stationary policy $\boldsymbol{\pi}$, the value function for an infinite-horizon discounted game is defined as,

$$V_k^{\boldsymbol{\pi}}(s_1) = \mathbb{E}_{\boldsymbol{\pi}}\left[\sum_{h=1}^{H} \gamma^{h-1} r_{k,h}(s_h, \boldsymbol{a}_h)\big| s_1\right] = \boldsymbol{e}_{s_1}^{\top} \sum_{h=1}^{H} \left(\gamma^{h-1} \prod_{\tau=1}^{h} \mathbb{P}_{\tau}(\boldsymbol{\pi}_{\tau})\right) \boldsymbol{r}_{k,h}(\boldsymbol{\pi}_h).$$

It is possible to express the value function of each player $k$ in the following way,

$$V_k^{\boldsymbol{\pi}}(s_1) = \boldsymbol{e}_{s_1}^{\top} \left(\mathbf{I} - \gamma \mathbb{P}(\boldsymbol{\pi})\right)^{-1} \boldsymbol{r}(\boldsymbol{\pi}).$$

Where $\mathbf{I}$ is the identity matrix of appropriate dimensions. Also, when the initial state is drawn from the initial state distribution, we denote, the value function reads $V_k^{\boldsymbol{\pi}}(\boldsymbol{\rho}) = \boldsymbol{\rho}^{\top} \left(\mathbf{I} - \gamma \mathbb{P}(\boldsymbol{\pi})\right)^{-1} \boldsymbol{r}(\boldsymbol{\pi})$.

**Best-response policies.**    Given an arbitrary joint policy $\boldsymbol{\sigma}$ (which can be either a correlated or product policy), a best-response policy of a player $k$ is defined to be $\boldsymbol{\pi}_k^{\dagger} \in \Delta(\mathcal{A}_k)^S$ such that $\boldsymbol{\pi}_k^{\dagger} \in \arg\max_{\boldsymbol{\pi}_k'} V_k^{\boldsymbol{\pi}_k' \times \boldsymbol{\sigma}_{-k}}(s)$. Also, we will denote $V_k^{\dagger, \boldsymbol{\sigma}_{-k}}(s) = \max_{\boldsymbol{\pi}_k'} V_k^{\boldsymbol{\pi}_k', \boldsymbol{\sigma}_{-k}}(s)$. It is rather straightforward to see that the problem of computing a best-response to a given policy is equivalent to solving a single-agent MDP problem.

**Notions of equilibria.** Now that best-response policies have been defined, it is straightforward to define the different notions of equilibria. First, we define the notion of a coarse-correlated equilibrium.

**Definition B.1** (CCE—infinite-horizon). *A joint (potentially correlated) policy $\boldsymbol{\sigma} \in \Delta(\mathcal{A})^S$ is an $\epsilon$-approximate coarse-correlated equilibrium if it holds that for an $\epsilon$,*

$$V_k^{\dagger, \boldsymbol{\sigma}_{-k}}(\boldsymbol{\rho}) - V_k^{\boldsymbol{\sigma}}(\boldsymbol{\rho}) \leq \epsilon, \ \forall k \in [n].$$

Second, we define the notion of a Nash equilibrium. The main difference of the definition of the coarse-correlated equilibrium, is the fact that a NE Markovian stationary policy is a *product policy*.

**Definition B.2** (NE—infinite-horizon). *A joint (potentially correlated) policy $\boldsymbol{\pi} \in \prod_{k \in [n]} \Delta(\mathcal{A}_k)^S$ is an $\epsilon$-approximate coarse-correlated equilibrium if it holds that for an $\epsilon$,*

$$V_k^{\dagger, \boldsymbol{\pi}_{-k}}(\boldsymbol{\rho}) - V_k^{\boldsymbol{\pi}}(\boldsymbol{\rho}) \leq \epsilon, \ \forall k \in [n].$$

As it is folklore by now, infinite-horizon discounted Markov games have a stationary Markovian Nash equilibrium.

## C  Main results for infinite-horizon games

The workhorse of our arguments in the following results is still the following nonlinear program with variables $\boldsymbol{\pi}, \boldsymbol{w}$,

$$
\begin{aligned}
\min \ & \sum_{k \in [n]} \boldsymbol{\rho}^\top \left( \boldsymbol{w}_k - (\mathbf{I} - \gamma \, \mathbb{P}(\boldsymbol{\pi}))^{-1} \boldsymbol{r}_k(\boldsymbol{\pi}) \right) \\
\text{s.t. } & w_k(s) \geq r_k(s, a, \boldsymbol{\pi}_{-k}) + \gamma \, \mathbb{P}(s, a, \boldsymbol{\pi}_{-k}) \boldsymbol{w}_k, \\
& \quad \forall s \in \mathcal{S}, \forall k \in [n], \forall a \in \mathcal{A}_k; \\
& \boldsymbol{\pi}_k(s) \in \Delta(\mathcal{A}_k), \\
& \quad \forall s \in \mathcal{S}, \forall k \in [n], \forall a \in \mathcal{A}_k.
\end{aligned}
$$

($\mathrm{P}'_{\mathrm{NE}}$)

As we will prove, approximate NE's correspond to approximate global minima of ($\mathrm{P}'_{\mathrm{NE}}$) and vice-versa. Before that, we need some intermediate lemmas. The first lemma we prove is about the best-response program.

**The best-response program.** Even for the infinite-horizon, we can define a linear program for the best-responses of all players. That program is the following, with variables $\boldsymbol{w}$,

$$
\begin{aligned}
\min \ & \sum_{k \in [n]} \boldsymbol{\rho}^\top \left( \boldsymbol{w}_k - (\mathbf{I} - \gamma \, \mathbb{P}(\hat{\boldsymbol{\sigma}}))^{-1} \boldsymbol{r}_k(\hat{\boldsymbol{\sigma}}) \right) \\
\text{s.t. } & w_k(s) \geq r_k(s, a, \hat{\boldsymbol{\sigma}}_{-k}) + \mathbb{P}(s, a, \hat{\boldsymbol{\sigma}}_{-k}) \boldsymbol{w}_k, \\
& \quad \forall s \in \mathcal{S}, \forall k \in [n], \forall a \in \mathcal{A}_k.
\end{aligned}
$$

($\mathrm{P}'_{\mathrm{BR}}$)

**Lemma C.1** (Best-response LP—infinite-horizon). *Let a (possibly correlated) joint policy $\hat{\boldsymbol{\sigma}}$. Consider the linear program ($\mathrm{P}'_{\mathrm{BR}}$). The optimal solution $\boldsymbol{w}^\dagger$ of the program is unique and corresponds to the value function of each player $k \in [n]$ when player $k$ best-responds to $\hat{\boldsymbol{\sigma}}$.*

**Proof.** We observe that the program is separable to $n$ independent linear programs, each with variables $\boldsymbol{w}_k \in \mathbb{R}^n$,

$$
\begin{aligned}
\min \ & \boldsymbol{\rho}^\top \boldsymbol{w}_k \\
\text{s.t. } & w_k(s) \geq r_k(s, a, \hat{\boldsymbol{\sigma}}_{-k}) + \gamma \, \mathbb{P}(s, a, \hat{\boldsymbol{\sigma}}_{-k}) \boldsymbol{w}_k, \\
& \quad \forall s \in \mathcal{S}, \forall a \in \mathcal{A}_k.
\end{aligned}
$$

Each of these linear programs describes the problem of a single agent MDP —that agent being $k$. It follows that the optimal $\boldsymbol{w}_k^\dagger$ for every program is unique (each program corresponds to a set of Bellman optimality equations). $\qquad\square$

**Properties of the NE program.** Second, we need to prove that the minimum value of the objective function of the program is nonnegative.

**Lemma C.2** (Feasibility of ($P'_{NE}$) and global optimum)**.** The nonlinear program ($P'_{NE}$) is feasible, has a nonnegative objective value, and its global minimum is equal to 0.

**Proof.** For the feasibility of the nonlinear program, we invoke the theorem of the existence of a Nash equilibrium. *i.e.*, let a NE product policy, $\boldsymbol{\pi}^\star$, and a vector $\boldsymbol{w}^\star \in \mathbb{R}^{n \times H \times S}$ such that $w_{k,s}^\star(s) = V_k^{\dagger, \boldsymbol{\pi}^\star_{-k}}(s), \ \forall k \in [n] \times \mathcal{S}$.

By Lemma C.1, we know that $(\boldsymbol{\pi}^\star, \boldsymbol{w}^\star)$ satisfies all the constraints of ($P'_{NE}$). Additionally, because $\boldsymbol{\pi}^\star$ is a NE, $V_k^{\boldsymbol{\pi}^\star}(\boldsymbol{\rho}) = V_k^{\dagger, \boldsymbol{\pi}^\star_{-k}}(\boldsymbol{\rho})$ for all $k \in [n]$. Observing that,

$$\boldsymbol{\rho}^\top \left( \boldsymbol{w}_k^\star - (\mathbf{I} - \gamma \, \mathbb{P}(\boldsymbol{\pi}^\star))^{-1} \boldsymbol{r}_k(\boldsymbol{\pi}^\star) \right) = V_k^{\dagger, \boldsymbol{\pi}^\star_{-k}}(\boldsymbol{\rho}) - V_k^{\boldsymbol{\pi}^\star}(\boldsymbol{\rho}) = 0,$$

concludes the argument that a NE attains an objective value equal to 0.

Continuing, we observe that due to (1) the objective function can be equivalently rewritten as,

$$\sum_{k \in [n]} \left( \boldsymbol{\rho}^\top \boldsymbol{w}_k - \boldsymbol{\rho}^\top (\mathbf{I} - \gamma \, \mathbb{P}(\boldsymbol{\pi}))^{-1} \boldsymbol{r}_k(\boldsymbol{\pi}) \right)$$

$$= \sum_{k \in [n]} \boldsymbol{\rho}^\top \boldsymbol{w}_k - \boldsymbol{\rho}^\top (\mathbf{I} - \gamma \, \mathbb{P}(\boldsymbol{\pi}))^{-1} \sum_{k \in [n]} \boldsymbol{r}_k(\boldsymbol{\pi}_h)$$

$$= \sum_{k \in [n]} \boldsymbol{\rho}^\top \boldsymbol{w}_k.$$

Next, we focus on the inequality constraint

$$w_k(s) \geq r_k(s, a, \boldsymbol{\pi}_{-k}) + \gamma \, \mathbb{P}(s, a, \boldsymbol{\pi}_{-k}) \boldsymbol{w}_k$$

which holds for all $s \in \mathcal{S}$, all players $k \in [n]$, and all $a \in \mathcal{A}_k$.

By summing over $a \in \mathcal{A}_k$ while multiplying each term with a corresponding coefficient $\pi_k(a|s)$, the display written in an equivalent element-wise vector inequality reads:

$$\boldsymbol{w}_k \geq \boldsymbol{r}_{k,h}(\boldsymbol{\pi}) + \gamma \, \mathbb{P}(\boldsymbol{\pi}) \boldsymbol{w}_k.$$

Finally, after consecutively substituting $\boldsymbol{w}_k$ with the element-wise lesser term $\boldsymbol{r}_k(\boldsymbol{\pi}) + \gamma \, \mathbb{P}(\boldsymbol{\pi}) \boldsymbol{w}_k$, we end up with the inequality:

$$\boldsymbol{w}_k \geq (\mathbf{I} - \gamma \, \mathbb{P}(\boldsymbol{\pi}))^{-1} \boldsymbol{r}_k(\boldsymbol{\pi}). \tag{9}$$

We note that $\mathbf{I} + \gamma \, \mathbb{P}(\boldsymbol{\pi}) + \gamma^2 \, \mathbb{P}^2(\boldsymbol{\pi}) + \cdots = (\mathbf{I} - \gamma \, \mathbb{P}(\boldsymbol{\pi}))^{-1}$.

Summing over $k$, it holds for the $s_1$-th entry of the inequality,

$$\sum_{k \in [n]} \boldsymbol{w}_k \geq \sum_{k \in [n]} (\mathbf{I} - \gamma \, \mathbb{P}(\boldsymbol{\pi}))^{-1} \boldsymbol{r}_k(\boldsymbol{\pi}) = (\mathbf{I} - \gamma \, \mathbb{P}(\boldsymbol{\pi}))^{-1} \sum_{k \in [n]} \boldsymbol{r}_k(\boldsymbol{\pi}) = 0.$$

Where the equality holds due to the zero-sum property, (1). $\qquad \square$

**Theorem C.1** (NE and global optima of ($P'_{NE}$)—infinite-horizon)**.** *If $(\boldsymbol{\pi}^\star, \boldsymbol{w}^\star)$ yields an $\epsilon$-approximate global minimum of ($P'_{NE}$), then $\boldsymbol{\pi}^\star$ is an $n\epsilon$-approximate NE of the infinite-horizon zero-sum polymatrix switching controller MG, $\Gamma$. Conversely, if $\boldsymbol{\pi}^\star$ is an $\epsilon$-approximate NE of the MG $\Gamma$ with corresponding value function vector $\boldsymbol{w}^\star$ such that $w_k^\star(s) = V_k^{\boldsymbol{\pi}^\star}(s) \forall (k, s) \in [n] \times \mathcal{S}$, then $(\boldsymbol{\pi}^\star, \boldsymbol{w}^\star)$ attains an $\epsilon$-approximate global minimum of ($P'_{NE}$).*

**Proof.**
**An approximate NE is an approximate global minimum.** We show that an $\epsilon$-approximate NE, $\boldsymbol{\pi}^\star$, achieves an $n\epsilon$-approximate global minimum of the program. Utilizing Lemma C.1 by setting $\boldsymbol{w}_k^\star = \mathbf{V}^{\dagger, \boldsymbol{\pi}^\star_{-k}}(\boldsymbol{\rho})$, feasibility , and the definition of an $\epsilon$-approximate NE we see that,

$$\sum_{k\in[n]}\left(\boldsymbol{\rho}^\top\boldsymbol{w}_k^\star-\boldsymbol{\rho}^\top\left(\mathbf{I}-\gamma\,\mathbb{P}(\boldsymbol{\pi}^\star)\right)^{-1}\boldsymbol{r}_k(\boldsymbol{\pi}^\star)\right)=\sum_{k\in[n]}\left(\boldsymbol{\rho}^\top\boldsymbol{w}_k^\star-V_k^{\boldsymbol{\pi}^\star}(\boldsymbol{\rho})\right)$$

$$\leq\sum_{k\in[n]}\epsilon=n\epsilon.$$

Indeed, this means that $\boldsymbol{\pi}^\star,\boldsymbol{w}^\star$ is an $n\epsilon$-approximate global minimizer of ($\mathrm{P}'_{\mathrm{NE}}$).

**An approximate global minimum is an approximate NE.** For this direction, we let a feasible $\epsilon$-approximate global minimizer of the program ($\mathrm{P}'_{\mathrm{NE}}$), $(\boldsymbol{\pi}^\star,\boldsymbol{w}^\star)$. Because a global minimum of the program is equal to 0, an $\epsilon$-approximate global optimum must be at most $\epsilon>0$. We observe that for every $k\in[n]$,

$$\boldsymbol{\rho}^\top\boldsymbol{w}_k^\star\geq\boldsymbol{\rho}^\top\left(\mathbf{I}-\gamma\,\mathbb{P}(\boldsymbol{\pi}^\star)\right)^{-1}\boldsymbol{r}_k(\boldsymbol{\pi}^\star),\tag{10}$$

which follows from induction on the inequality constraint (9).

Consequently, the assumption that

$$\epsilon\geq\boldsymbol{\rho}^\top\boldsymbol{w}_k^\star-\boldsymbol{\rho}^\top\left(\mathbf{I}-\gamma\,\mathbb{P}(\boldsymbol{\pi}^\star)\right)^{-1}\boldsymbol{r}_k(\boldsymbol{\pi}^\star)$$

and Equation (10), yields the fact that

$$\epsilon\geq\boldsymbol{\rho}^\top\boldsymbol{w}_k^\star-\boldsymbol{\rho}^\top\left(\mathbf{I}-\gamma\,\mathbb{P}(\boldsymbol{\pi}^\star)\right)^{-1}\boldsymbol{r}_k(\boldsymbol{\pi}^\star)$$
$$\geq V_k^{\dagger,\boldsymbol{\pi}^\star_{-k}}(\boldsymbol{\rho})-V_k^{\boldsymbol{\pi}^\star}(\boldsymbol{\rho}),$$

where the second inequality holds from the fact that $\boldsymbol{w}^\star$ is also feasible for ($\mathrm{P}'_{\mathrm{BR}}$). The latter concludes the proof, as the display coincides with the definition of an $\epsilon$-approximate NE. $\qquad\square$

**Theorem C.2** (CCE collapse to NE in polymatrix MG—infinite-horizon). *Let a zero-sum polymatrix switching-control Markov game, i.e., a Markov game for which Assumptions 1 and 2 hold. Further, let an $\epsilon$-approximate CCE of that game $\boldsymbol{\sigma}$. Then, the marginal product policy $\boldsymbol{\pi}^\sigma$, with $\pi_k^\sigma(a|s)=\sum_{\boldsymbol{a}_{-k}\in\mathcal{A}_{-k}}\boldsymbol{\sigma}(a,\boldsymbol{a}_{-k})$, $\forall k\in[n]$ is an $n\epsilon$-approximate NE.*

**Proof.** Let an $\epsilon$-approximate CCE policy, $\boldsymbol{\sigma}$, of game $\Gamma$. Moreover, let the best-response value-vectors of each agent $k$ to joint policy $\boldsymbol{\sigma}_{-k}$, $\boldsymbol{w}_k^\dagger$.

Now, we observe that due to Assumption 1,

$$w_k^\dagger(s)\geq r_k(s,a,\boldsymbol{\sigma}_{-k})+\gamma\,\mathbb{P}_h(s,a,\boldsymbol{\sigma}_{-k})\boldsymbol{w}_k^\dagger$$
$$=\sum_{j\in\mathrm{adj}(k)}r_{(k,j),h}(s,a,\boldsymbol{\pi}_j^\sigma)+\gamma\,\mathbb{P}(s,a,\boldsymbol{\sigma}_{-k})\boldsymbol{w}_k^\dagger.$$

Further, due to Assumption 2,

$$\mathbb{P}(s,a,\boldsymbol{\sigma}_{-k})\boldsymbol{w}_k^\dagger=\mathbb{P}(s,a,\boldsymbol{\pi}_{\mathrm{argctrl}(s)}^\sigma)\boldsymbol{w}_k^\dagger,$$

or,

$$\mathbb{P}(s,a,\boldsymbol{\sigma}_{-k})\boldsymbol{w}_k^\dagger=\mathbb{P}(s,a,\boldsymbol{\pi}^\sigma)\boldsymbol{w}_k^\dagger.$$

Putting these pieces together, we reach the conclusion that $(\boldsymbol{\pi}^\sigma,\boldsymbol{w}^\dagger)$ is feasible for the nonlinear program ($\mathrm{P}'_{\mathrm{NE}}$).

What is left is to prove that it is also an $\epsilon$-approximate global minimum. Indeed, if $\sum_k\boldsymbol{\rho}^\top\boldsymbol{w}_k^\dagger\leq\epsilon$ (by assumption of an $\epsilon$-approximate CCE), then the objective function of ($\mathrm{P}'_{\mathrm{NE}}$) will attain an $\epsilon$-approximate global minimum. In turn, due to Theorem C.1 the latter implies that $\boldsymbol{\pi}^\sigma$ is an $n\epsilon$-approximate NE. $\qquad\square$

## C.1 No equilibrium collapse with more than one controllers per-state

**Example 2.** *We consider the following 3-player Markov game that takes place for a time horizon $H=3$. There exist three states, $s_1,s_2,$ and $s_3$ and the game starts at state $s_1$. Player 3 has a single action in every state, while players 1 and 2 have two available actions $\{a_1,a_2\}$ and $\{b_1,b_2\}$ respectively in every state. The initial state distribution $\boldsymbol{\rho}$ is the uniform probability distribution over $\mathcal{S}$.*

**Reward functions.** *If player 1 (respectively, player 2) takes action $a_1$ (resp., $b_1$), in either of the states $s_1$ or $s_2$, they get a reward equal to $\frac{1}{20}$. In state $s_3$, both players get a reward equal to $-\frac{1}{2}$ regardless of the action they select. Player 3 always gets a reward that is equal to the negative sum of the reward of the other two players. This way, the* zero-sum polymatrix property *of the game is ensured (Assumption 1).*

**Transition probabilities.** *If players 1 and 2 select the joint action $(a_1, b_1)$ in state $s_1$, the game will transition to state $s_2$. In any other case, it will transition to state $s_3$. The converse happens if in state $s_2$ they take joint action $(a_1, b_1)$; the game will transition to state $s_3$. For any other joint action, it will transition to state $s_1$. From state $s_3$, the game transition to state $s_1$ or $s_2$ uniformly at random.*

*At this point, it is important to notice that two players control the transition probability from one state to another. In other words, Assumption 2 does not hold.*

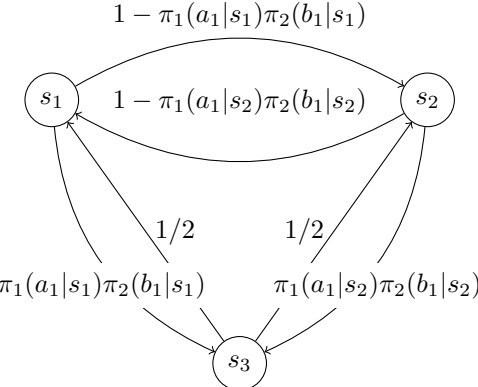

Figure 2: A graph of the state space with transition probabilities parametrized with respect to the policy of each player.

*Next, we consider the joint policy $\boldsymbol{\sigma}$,*

$$
\boldsymbol{\sigma}(s_1) = \boldsymbol{\sigma}(s_2) = \begin{array}{c} \\ a_1 \\ a_2 \end{array} \begin{pmatrix} \overset{b_1}{0} & \overset{b_2}{1/2} \\ 1/2 & 0 \end{pmatrix}.
$$

**Claim C.1.** The joint policy $\boldsymbol{\sigma}$ that assigns probability $\frac{1}{2}$ to the joint actions $(a_1, b_2)$ and $(a_2, b_1)$ in both states $s_1, s_2$ is a CCE and $V_1^{\boldsymbol{\sigma}}(\boldsymbol{\rho}) = V_2^{\boldsymbol{\sigma}}(\boldsymbol{\rho}) = -\frac{1}{10}$.

**Proof.**

$$
\begin{aligned}
V_1^{\boldsymbol{\sigma}}(\boldsymbol{\rho}) &= \boldsymbol{\rho}^\top \left( \mathbf{I} - \gamma \, \mathbb{P}(\boldsymbol{\sigma}) \right)^{-1} \boldsymbol{r}_1(\boldsymbol{\sigma}) \\
&= \begin{pmatrix} \frac{1}{3} & \frac{1}{3} & \frac{1}{3} \end{pmatrix} \begin{pmatrix} \frac{9}{5} & \frac{6}{5} & 0 \\ \frac{6}{5} & \frac{9}{5} & 0 \\ 1 & 1 & 1 \end{pmatrix} \begin{pmatrix} \frac{1}{40} \\ \frac{1}{40} \\ -\frac{1}{2} \end{pmatrix} \\
&= -\frac{1}{10}.
\end{aligned}
$$

We check every deviation,

- $\boldsymbol{\pi}_1(s_1) = \boldsymbol{\pi}_1(s_2) = \begin{pmatrix} 1 & 0 \end{pmatrix}, V^{\boldsymbol{\pi}_1 \times \boldsymbol{\sigma}_{-1}}(\boldsymbol{\rho}) = -\frac{2}{5}$,

- $\boldsymbol{\pi}_1(s_1) = \boldsymbol{\pi}_1(s_2) = \begin{pmatrix} 0 & 1 \end{pmatrix}, V^{\boldsymbol{\pi}_1 \times \boldsymbol{\sigma}_{-1}}(\boldsymbol{\rho}) = -\frac{1}{6}$,

- $\boldsymbol{\pi}_1(s_1) = \begin{pmatrix} 1 & 0 \end{pmatrix}, \boldsymbol{\pi}_1(s_2) = \begin{pmatrix} 0 & 1 \end{pmatrix}, V^{\boldsymbol{\pi}_1 \times \boldsymbol{\sigma}_{-1}}(\boldsymbol{\rho}) = -\frac{5}{16}$,

- $\boldsymbol{\pi}_1(s_1) = \begin{pmatrix} 0 & 1 \end{pmatrix}, \boldsymbol{\pi}_1(s_2) = \begin{pmatrix} 1 & 0 \end{pmatrix}, V^{\boldsymbol{\pi}_1 \times \boldsymbol{\sigma}_{-1}}(\boldsymbol{\rho}) = -\frac{5}{16}$.

For every such deviation the value of player 1 is smaller than $-\frac{1}{10}$. For player 2, the same follows by symmetry. Hence, $\boldsymbol{\sigma}$ is indeed a CCE.

$\square$

*Yet, the marginalized product policy of $\boldsymbol{\sigma}$ which we note as $\boldsymbol{\pi}_1^{\boldsymbol{\sigma}} \times \boldsymbol{\pi}_2^{\boldsymbol{\sigma}}$ does not constitute a NE. The components of this policy are,*

$$
\begin{cases}
\boldsymbol{\pi}_1^{\boldsymbol{\sigma}}(s_1) = \boldsymbol{\pi}_1^{\boldsymbol{\sigma}}(s_2) = \begin{pmatrix} \overset{a_1}{1/2} & \overset{a_2}{1/2} \end{pmatrix}, \\[2ex]
\boldsymbol{\pi}_2^{\boldsymbol{\sigma}}(s_1) = \boldsymbol{\pi}_2^{\boldsymbol{\sigma}}(s_2) = \begin{pmatrix} \overset{b_1}{1/2} & \overset{b_2}{1/2} \end{pmatrix}.
\end{cases}
$$

*I.e., the product policy $\boldsymbol{\pi}_1^{\boldsymbol{\sigma}} \times \boldsymbol{\pi}_2^{\boldsymbol{\sigma}}$ selects any of the two actions of each player in states $s_1, s_2$ independently and uniformly at random. With the following claim, it can be concluded that in general when more than one player control the transition the set of equilibria do not collapse.*

**Claim C.2.** The product policy $\boldsymbol{\pi}_1^{\boldsymbol{\sigma}} \times \boldsymbol{\pi}_2^{\boldsymbol{\sigma}}$ is not a NE.

**Proof.** For $\boldsymbol{\pi}^{\sigma} = \boldsymbol{\pi}_1^{\boldsymbol{\sigma}} \times \boldsymbol{\pi}_2^{\boldsymbol{\sigma}}$ we get,

$$
\begin{aligned}
V_1^{\boldsymbol{\pi}^{\sigma}} &= \boldsymbol{\rho}^{\top} \left(\mathbf{I} - \gamma \, \mathbb{P}(\boldsymbol{\pi}^{\boldsymbol{\sigma}})\right)^{-1} \boldsymbol{r}_1(\boldsymbol{\pi}^{\boldsymbol{\sigma}}) \\
&= \begin{pmatrix} \frac{1}{3} & \frac{1}{3} & \frac{1}{3} \end{pmatrix} \begin{pmatrix} \frac{34}{21} & \frac{20}{21} & \frac{3}{7} \\ \frac{20}{21} & \frac{34}{21} & \frac{3}{7} \\ \frac{6}{7} & \frac{6}{7} & \frac{9}{7} \end{pmatrix} \begin{pmatrix} \frac{1}{40} \\ \frac{1}{40} \\ -\frac{1}{2} \end{pmatrix} \\
&= -\frac{3}{10}.
\end{aligned}
$$

But, for the deviation $\boldsymbol{\pi}_1(a_1|s_1) = \boldsymbol{\pi}_1(a_1|s_2) = 0$, the value funciton of player 1, is equal to $-\frac{1}{6}$. Hence, $\boldsymbol{\pi}^{\sigma}$ is not a NE. $\square$

*In conclusion, Assumption 1 does not suffice to ensure equilibrium collapse.*

**Theorem C.3** (No collapse—infinite-horizon)**.** *There exists a zero-sum polymatrix Markov game (Assumption 2 is not satisfied) that has a CCE which does not collapse to a NE.*

**Proof.** The proof follows from the game of Example 2, and Claims C.1 and C.2. $\square$

# D An algorithm for approximating Markovian CCE

In this section we describe the algorithm in (Daskalakis et al., 2022) used for the computation a of Markovian CCE. We note that $M, N_{\text{visit}}, p$ are parameters that affect the accuracy of the approximation and we kindly ask the reader to refer to (Daskalakis et al., 2022) for further details. This algorithm computes an $\epsilon$-approximate CCE in time $\tilde{O}(\epsilon^{-3})$. Newer works have improved the dependence on $\epsilon$ to $\tilde{O}(\epsilon^{-2})$, see (Wang et al., 2023; Cui et al., 2023).

As soon as an $\epsilon$-approximate CCE is computed, what is left to do is to compute the marginal policy of every player, $\pi_{k,h}^{\boldsymbol{\sigma}}(a|s) = \sum_{\boldsymbol{a}_{-k} \in \mathcal{A}_{-k}} \sigma(a, \boldsymbol{a}_{-k}|s)$.

**Algorithm 1** SPoCMAR (Daskalakis et al., 2022)

1: **procedure** SPoCMAR($n, \mathcal{S}, \mathcal{A}, H, M, N_{\text{visit}}, p$)
2:      Set $\mathcal{V} = \emptyset$.
3:      For each $h \in [H]$, $s \in \mathcal{S}$, set $\boldsymbol{\sigma}_{h,s}^{\text{cover}} = \perp$.
4:      **for** $q \geq 1$ and while $\tau = 0$ **do**
5:          Set $\tau = 1$                                      `#Terminate flag.`
6:          Set $\Pi_h^q := \{\boldsymbol{\sigma}_{h,s}^{\text{cover}} : s \in \mathcal{S}\}$ for each $h \in [H]$.
7:          **for** $h = H, H-1, \ldots, 1$ **do**
8:              Set $k = 0$, and $\bar{V}_{i,H+1}^q(s) = 0$ for all $s \in \mathcal{S}$ and $i \in [m]$.
9:              Each player $i$ initializes an adversarial bandit for all $(s, h) \in \mathcal{S} \times [H]$.
10:             **for** each $\boldsymbol{\sigma} \in \Pi_h^q \cup \boldsymbol{\sigma}^{\mathcal{U}}$ **do**     `# `$\boldsymbol{\sigma}^{\mathcal{U}}$` chooses actions uniformly at random`
11:                 **for** a total of $M$ times **do**
12:                     $k \leftarrow k + 1$.
13:                     Let $\bar{\boldsymbol{\sigma}}$ be the policy which follows $\boldsymbol{\sigma}$ for the first $h-1$ steps and plays according to the bandit algorithm for the state visited at step $h$ (and acts arbitrarily for steps $h' > h$).
14:                     Draw a joint trajectory $(s_{1,m}, \boldsymbol{a}_{1,m}, \boldsymbol{r}_{1,m}, \ldots, s_{H,m}, \boldsymbol{a}_{H,m}, \boldsymbol{r}_{H,m})$ from $\bar{\boldsymbol{\sigma}}$.
15:                     **if** $(h, s_{h,m}) \in \mathcal{V}$ **then**
16:                         Each $i$ updates its bandit alg. at $(h, s_{h,m})$ with $\left(a_{i,h,m}, \frac{H - r_{i,h,m} - \bar{V}_{i,h+1}^q(s_{h+1,m})}{H}\right)$.
17:                     **else**
18:                         Each $i$ updates its bandit alg. at $(h, s_{h,m})$ with $\left(a_{i,h,m}, \frac{H - (H+1-h)}{H}\right)$.
19:                     **end if**
20:                 **end for**
21:             **end for**
22:             For each $s \in \mathcal{S}$, and $j \geq 1$, let $m_{j,h,s} \in [M+1]$ denote the $j$th smallest value of $k$ so that $s_{h,m} = s$, or $M + 1$ if such a $j$th smallest value does not exist.
23:             For each $s \in \mathcal{S}$, let $J_{h,s}$ denote the largest integer $j$ so that $k_{j,h,s} \leq M$.
24:             Define $\tilde{\boldsymbol{\sigma}}_h^q \in \Delta(\mathcal{A})^{\mathcal{S}}$ to be the 1-step policy: $\tilde{\boldsymbol{\sigma}}_h^q(\boldsymbol{a}|s) = \frac{1}{J_{h,s}} \sum_{j=1}^{J_{h,s}} \mathbb{1}[\boldsymbol{a} = \boldsymbol{a}_{h,m_{j,h,s}}]$.
25:             Set

$$\bar{V}_{i,h}^q(s) := \begin{cases} \frac{1}{J_{h,s}} \sum_{j=1}^{J_{h,s}} \left( r_{i,h,k_{j,h,s}} + \bar{V}_{i,h+1}^q(s_{h+1,k_{j,h,s}}) \right) & : (h,s) \in \mathcal{V} \\ (H+1-h) & : (h,s) \notin \mathcal{V}. \end{cases}$$

26:          **end for**
27:          Define the joint policy $\tilde{\boldsymbol{\sigma}}^q$, which follows $\tilde{\boldsymbol{\sigma}}_{h'}^q$ at each step $h' \in [H]$.
28:          Call EstVisitation($\tilde{\boldsymbol{\sigma}}^q, N_{\text{visit}}$) (Alg. 2) to obtain estimates $\hat{d}_{h'}^q \in \Delta(\mathcal{S})$ for each $h' \in [H]$.
29:          **for** each $s \in \mathcal{S}$ and $h' \in [H]$ **do**
30:             **if** $\hat{d}_{h'}^q(s) \geq p$ and $(h', s) \notin \mathcal{V}$ **then**
31:                 Set $\boldsymbol{\sigma}_{h',s}^{\text{cover}} \leftarrow \tilde{\boldsymbol{\sigma}}^q$.
32:                 Add $(h', s)$ to $\mathcal{V}$.
33:                 Set $\tau \leftarrow 0$.
34:             **end if**
35:          **end for**
36:      **end for**
37:      **return** the policy $\hat{\boldsymbol{\sigma}} := \tilde{\boldsymbol{\sigma}}^q$.
38: **end procedure**

---

**Algorithm 2** EstVisitation

---

1: **procedure** EstVisitation($\boldsymbol{\sigma}, N$)
2:     **for** $1 \leq n \leq N$ **do**
3:         Draw a trajectory from $\boldsymbol{\sigma}$, and let $(s_1^n, \ldots, s_H^n)$ denote the sequence of states observed.
4:     **end for**
5:     **for** $h \in [H]$ **do**
6:         Let $\hat{d}_h \in \Delta(\mathcal{S})$ denote the empirical distribution over $(s_h^1, \ldots, s_h^N)$.
7:     **end for**
8:     **return** $(\hat{d}_1, \ldots, \hat{d}_H)$.
9: **end procedure**

---

