## A  Missing statements and proofs

### A.1  Statements for Section 3.1

**Claim A.1.** Let a two-player Markov game where both players affect the transition. Further, consider a correlated policy $\boldsymbol{\sigma}$ and its corresponding marginalized product policy $\boldsymbol{\pi}^{\boldsymbol{\sigma}} = \boldsymbol{\pi}_1^{\boldsymbol{\sigma}} \times \boldsymbol{\pi}_2^{\boldsymbol{\sigma}}$. Then, for any $\boldsymbol{\pi}_1', \boldsymbol{\pi}_2'$,

$$V_{k,1}^{\boldsymbol{\pi}_1', \boldsymbol{\sigma}^{-1}}(s_1) = V_{k,1}^{\boldsymbol{\pi}_1', \boldsymbol{\pi}_2^{\boldsymbol{\sigma}}}(s_1),$$

$$V_{k,2}^{\boldsymbol{\sigma}^{-2}, \boldsymbol{\pi}_2'}(s_1) = V_{k,2}^{\boldsymbol{\pi}_1^{\boldsymbol{\sigma}}, \boldsymbol{\pi}_2'}(s_1).$$

**Proof.** We will effectively show that the problem of best-responding to a correlated policy $\boldsymbol{\sigma}$ is equivalent to best-responding to the marginal policy of $\boldsymbol{\sigma}$ for the opponent. The proof follows from the equivalence of the two MDPs.

As a reminder,

$$\pi_{1,h}(a|s) = \sum_{b \in \mathcal{A}_2} \boldsymbol{\sigma}_h(a,b|s)$$

$$\pi_{2,h}(b|s) = \sum_{a \in \mathcal{A}_1} \boldsymbol{\sigma}_h(a,b|s)$$

As we have seen in Section 2.1, in the case of unilateral deviation from joint policy $\boldsymbol{\sigma}$, an agent faces a single agent MDP. More specifically, agent 2, best-responds by optimizing a reward function $\bar{r}_{2,h}(s,b)$ under a transition kernel $\bar{\mathbb{P}}_2$ for which,

$$\bar{r}_{2,h}(s,b) = \mathbb{E}_{b \sim \boldsymbol{\sigma}}\left[r_{2,h}(s,a,b)\right] = \mathbb{E}_{b \sim \boldsymbol{\pi}_1^{\boldsymbol{\sigma}}}\left[r_{2,h}(s,a,b)\right] = r_{2,h}(s,\boldsymbol{\pi}_1^{\boldsymbol{\sigma}},b).$$

Similarly,

$$\bar{r}_{1,h}(s,b) = r_{1,h}(s,a,\boldsymbol{\pi}_2^{\boldsymbol{\sigma}}).$$

Analogously, for each of the transition kernels,

$$\bar{\mathbb{P}}_{2,h}(s'|s,b) = \mathbb{E}_{a \sim \boldsymbol{\sigma}}\left[\mathbb{P}_{2,h}(s'|s,a,b)\right] = \mathbb{E}_{a \sim \boldsymbol{\pi}_2^{\boldsymbol{\sigma}}}\left[\mathbb{P}_{2,h}(s'|s,a,b)\right] = \mathbb{P}_{2,h}(s'|s,\boldsymbol{\pi}_1^{\boldsymbol{\sigma}},b),$$

as for agent 1,

$$\bar{\mathbb{P}}_{1,h}(s'|s,a) = \mathbb{P}_{1,h}(s'|s,a,\boldsymbol{\pi}_2^{\boldsymbol{\sigma}}).$$

Hence, it follows that, $V_{2,1}^{\boldsymbol{\sigma}^{-2} \times \boldsymbol{\pi}_2'}(s_1) = V_{2,1}^{\boldsymbol{\pi}_1^{\boldsymbol{\sigma}} \times \boldsymbol{\pi}_2'}(s_1), \ \forall \boldsymbol{\pi}_2'$ and $V_{1,1}^{\boldsymbol{\pi}_1' \times \boldsymbol{\sigma}^{-1}}(s_1) = V_{1,1}^{\boldsymbol{\pi}_1' \times \boldsymbol{\pi}_2^{\boldsymbol{\sigma}}}(s_1), \ \forall \boldsymbol{\pi}_2'$.

$\square$

Before that, given a (possibly correlated) joint policy $\boldsymbol{\sigma}$ we define a nonlinear program, ($\mathrm{P_{BR}}$), whose optimal solutions are best-response policies of each agent $k$ to $\boldsymbol{\sigma}_{-k}$ and the values for each state $s$ and timestep $h$:

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

 $\pi^\star$ is a NE, $V_k^{\pi^\star}(\rho) = V_k^{\dagger,\pi^\star_{-k}}(\rho)$ for all $k \in [n]$. Observing that,

$$\rho^\top \left(w^\star_k - (\mathbf{I} - \gamma\,\mathbb{P}(\pi^\star))^{-1} r_k(\pi^\star)\right) = V_k^{\dagger,\pi^\star_{-k}}(\rho) - V_k^{\pi^\star}(\rho) = 0,$$

concludes the argument that a NE attains an objective value equal to 0.

Continuing, we observe that due to (1) the objective function can be equivalently rewritten as,

$$\sum_{k\in[n]} \left(\rho^\top w_k - \rho^\top (\mathbf{I} - \gamma\,\mathbb{P}(\pi))^{-1} r_k(\pi)\right)$$
$$= \sum_{k\in[n]} \rho^\top w_k - \rho^\top (\mathbf{I} - \gamma\,\mathbb{P}(\pi))^{-1} \sum_{k\in[n]} r_k(\pi_h)$$
$$= \sum_{k\in[n]} \rho^\top w_k.$$

Next, we focus on the inequality constraint

$$w_k(s) \ge r_k(s, a, \pi_{-k}) + \gamma\,\mathbb{P}(s, a, \pi_{-k}) w_k$$

which holds for all $s \in \mathcal{S}$, all players $k \in [n]$, and all $a \in \mathcal{A}_k$.

By summing over $a \in \mathcal{A}_k$ while multiplying each term with a corresponding coefficient $\pi_k(a|s)$, the display written in an equivalent element-wise vector inequality reads:

$$w_k \ge r_{k,h}(\pi) + \gamma\,\mathbb{P}(\pi) w_k.$$

Finally, after consecutively substituting $w_k$ with the element-wise lesser term $r_k(\pi) + \gamma\,\mathbb{P}(\pi) w_k$, we end up with the inequality:

$$w_k \ge (\mathbf{I} - \gamma\,\mathbb{P}(\pi))^{-1} r_k(\pi). \tag{9}$$

We note that $\mathbf{I} + \gamma\,\mathbb{P}(\pi) + \gamma^2\,\mathbb{P}^2(\pi) + \cdots = (\mathbf{I} - \gamma\,\mathbb{P}(\pi))^{-1}$.

Summing over $k$, it holds for the $s_1$-th entry of the inequality,

$$\sum_{k\in[n]} w_k \ge \sum_{k\in[n]} (\mathbf{I} - \gamma\,\mathbb{P}(\pi))^{-1} r_k(\pi) = (\mathbf{I} - \gamma\,\mathbb{P}(\pi))^{-1} \sum_{k\in[n]} r_k(\