# OpenReview forum: "Zero-sum Polymatrix Markov Games: Equilibrium Collapse and Efficient Computation of Nash Equilibria"
_NeurIPS.cc/2023/Conference — NeurIPS 2023 poster_

### Official Review · Reviewer_u7Ge · 2023-07-03

**Soundness:** 4 excellent
**Presentation:** 3 good
**Contribution:** 3 good
**Rating:** 7
**Confidence:** 4

**Summary:**

Cai (2016) proved that in normal-form polymatrix games the set of CCEs corresponds with the set of NEs. This paper provides a similar result for polymatrix, switching controller, Markov games.

After introducing the formalism used throughout the paper, the main result is presented for both fixed horizon and infinite horizon. A counter example is also offered regarding the necessity of the switching controller condition.

The crucial aspect of the paper from a theoretical perspective lies in the fact that the polymatrix + switching controller assumptions allow to switch from a correlated distribution to a product distribution in the formulation of a CCE, thus proving the equivalence to a NE.

**Strengths:**

- Clear step by step structure of the paper
- Solid and intuitive proofs

**Weaknesses:**

The clarity of the paper can be improved in multiple points:
1. Sometimes the notation is confusing and feels heavy
	- the $\cdot^\dagger$ for best responses is unintuitive and does not feel natural
	- appendix A.1 is missing a cartesian product $\times$ in the Best responses symbols (after line 525)
	- in many points of the paper, some symbols are silently redefined to avoid explicit expectation terms, like $r_{k,h}(s,a,b)$ suddenly accepting a probability distribution as in $r_{k,h}(s, \pi, b)$. At a first glance, this formalism bugged me. Having those shortcuts defined in line 167-169 would help in avoiding surprises in the formalism.
 2. The definition of the linear program $P_{NE}$ would be easier to comprehend if the program was accompanied by a short textual description of the variables' meaning and of the constraints. This will greatly improve the smoothness of the read and ease the comprehension of the proofs.
 3. The relation between CCEs and Best response policies defined as product distribution should be explicitly explained to allow the connection between the definition of the coarse correlated and its practical meaning (i.e. CCEs are stable to policy deviations which renounce to see the correlation signal). It is to be noted that at the moment there is no proper intuition behind the idea of a CCE.

**Questions:**

After carefully reading the paper, I have some questions/comments regarding specific passages of the paper:
1. I suggest to move lines 167-169 *before* the use of the defined symbols
2. Why is the warm-up in section 3.1 included in the body of the paper? I struggle to see its usefulness to make the whole paper clearer
3. Is $w_k^\dagger$ missing a $\cdot_h$ subscript in line 255?
4. is the sole purpose if including the subtraction of the expected value in the objective function of $P_{NE}$ to have a more comfortable global minimum in 0? (i.e. it simplifies the following statements)
5. Is the first constraint of program $P'_{NE}$ missing a $\gamma$ term?

Moreover, I'd like the authors to properly address the weaknesses from the previous sections.

On a side note, I highlight some of the typos:
- Shapely citation in line 18 seems to have the wrong format
- Unfinished sentence on line 72-73

Regarding the novelty of the technical approach and the relevance of the result, I think that those are good in the present paper, but I cannot evaluate them with high confidence as I work in a different yet related subfield. On the other hand, I checked the proofs in the main body and in the appendices A.1 and A.2 and I found them both clear and correct.

**Limitations:**

The main limitation of the paper is that specific assumptions have to be made on the reward structure (polymatrix) and transition function (switching control). These are clearly addressed and explicated in the paper. However, I do not agree with the authors regarding the practical real-world importance of polymatrix games (Section 1.1)

---

> ### Author Rebuttal · Authors · 2023-08-08
>
> We thank the reviewer for their comments and positive evaluation of our work. We commit to improving our text using the reviewer's insightful and helpful suggestions. We will try to answer the reviewer's concerns one by one. Please also see the global rebuttal response for further discussion of the mathematical program.
>
>
>
> >* Sometimes the notation is confusing and feels heavy
> > * the $\dagger$ for best responses is unintuitive and does not feel natural
>
> We used the $\dagger$ as the best-response symbol following some contermporary impactful papers [1], [2]. We struggled with how to make the notation lighter but it seems that MARL literature cannot circumvent a heavy notation -- of course, we want to know whether you have suggestions.
>
>
> > * At a first glance, this formalism bugged me. Having those shortcuts defined in line 167-169 would help in avoiding surprises in the formalism.
>
> Thank you for the corrections and suggestions. We will include all these shortcuts in the revised version.
>
> > The definition of the linear program would be easier to comprehend if the program was accompanied by a short textual description of the variables' meaning and of the constraints. This will greatly improve the smoothness of the read and ease the comprehension of the proofs. The relation between CCEs and Best response policies defined as product distribution should be explicitly explained to allow the connection between the definition of the coarse correlated and its practical meaning (i.e. CCEs are stable to policy deviations which renounce to see the correlation signal). It is to be noted that at the moment there is no proper intuition behind the idea of a CCE.
>
> Again, thank you for bringing this important oversight to our attention. Surely, we will need to add a relevant paragraph in our revised version.
>
> >After carefully reading the paper, I have some questions/comments regarding specific passages of the paper: I suggest to move lines 167-169 before the use of the defined symbols
>
> > Why is the warm-up in section 3.1 included in the body of the paper? I struggle to see its usefulness to make the whole paper clearer
>
> We think that it was important to include it for two reasons:
> * familiarize the readers with the notion of equilibrium collapse and a part of the arguments we later use,
> * our definition of switching control polymatrix zero-sum Markov games does not include two-player zero-sum Markov games where both players jointly control the transition function; yet, the same property of equilibrium collapse holds. (Please also see the observation we make
> in the rebuttal for reviewer zxYQ that polymatrix zero-sum MG's are PPAD with more than 2 players and more than one controller.)
>
> > is the sole purpose if including the subtraction of the expected value in the objective function of $P_{NE}$ to have a more comfortable global minimum in 0? (i.e. it simplifies the following statements)
>
> The answer is no. Intuitively, the objective function of the program is the sum of the gains when best-responding to a joint strategy profile $\mathbf{x}$. The gain is the utility when best-responding minus the current utility. Of course, we should add a note explaining the different parts of the program.
>
> In the case of zero-sum games, that quantity is zero as summing over all utilities is equal to zero.
>
> We refer to the global rebuttal text and [3] for a longer discussion of those mathematical programs.
>
>
> > Is the first constraint of program $P_{NE}'$ missing a $\gamma$ term?
> Yes, thanks for bringing this to our attention.
>
> > Moreover, I'd like the authors to properly address the weaknesses from the previous sections.
>
> Your comments have been very helpful and we will surely address them in our revised version.
>
>
> > The main limitation of the paper is that specific assumptions have to be made on the reward structure (polymatrix) and transition function (switching control). These are clearly addressed and explicated in the paper. However, I do not agree with the authors regarding the practical real-world importance of polymatrix games (Section 1.1)
>
> You are right that we are making specific assumptions. But, recent results ([1], [2])  have shown PPAD-hardness even for the computation of the relaxed notions of (Markov stationary) CCE in two-player general-sum Markov games. I.e., under well-founded computational complexities conjectures, it is unlikely there exists an efficient algorithm for their computation in general games. Hence, one of the main threads of MARL research is investigating settings amenable to efficient equilibrium computation. Our setting is of them.
>
> Nevertheless, as we mention to another reviewer, any transition rule that depends has the property of depending on the marginals of a correlated policy would work to ensure that the following equality holds:
> $$ P(s'|s, a, \sigma_{-k}) = P(s'|s, a, \pi^\sigma_{-k}) .$$
>
> Such a rule could be picking a controller at random at each state.
>
> ---
>
> Thanks,
>
> The authors
>
> ---
>
> [1] Jin, C., Liu, Q., Wang, Y. and Yu, T., 2021. V-Learning--A Simple, Efficient, Decentralized Algorithm for Multiagent RL. arXiv preprint arXiv:2110.14555.
>
> [2] Daskalakis, C., Golowich, N. and Zhang, K., 2023, July. The complexity of markov equilibrium in stochastic games. In The Thirty Sixth Annual Conference on Learning Theory (pp. 4180-4234). PMLR.
>
> [3] Filar, J.A., Schultz, T.A., Thuijsman, F. and Vrieze, O.J., 1991. Nonlinear programming and stationary equilibria in stochastic games. Mathematical Programming, 50(1-3), pp.227-237.

---

> > ### Comment · Reviewer_u7Ge · 2023-08-18
> >
> > The authors'rebuttal properly addresses my concerns. I confirm my original scoring of the paper

---

### Official Review · Reviewer_zxYQ · 2023-07-05

**Soundness:** 3 good
**Presentation:** 3 good
**Contribution:** 3 good
**Rating:** 6
**Confidence:** 4

**Summary:**

This paper defines a class of mulit-agent Markov games called zero-sum polymatrix Markov games, which is a generalization of the zero-sum polymatrix normal-form games. Specifically, it defines a class of Markov games where each state is a zero-sum polymatrix game. The main results of the paper shows that: in both the finite-horizon and the infinite-horizon setting (1) when the switching control assumption holds, i.e., the transition is determined by one player on each state, then the marginal policy of any (approximate) coerce correlated equilibrium (CCE) is an (approximate) Nash equilibrium (NE). This equilibrium collapse results ensures efficient computation of NE by reduction to computation of CCE. (2) when the transition is controlled by more than two players, equilibrium collapse does not hold.

**Strengths:**

1. This paper introduces an interesting class of Markov games where efficient computation of NE is possible, which covers switching control zero-sum Markov games and zero-sum polymatrix games as special cases. The results contribute to a better understanding of equilibrium computation in Markov games.
2. This paper is fairly well-written and easy to follow. The results are complete in the sense that the authors showed equilibrium collapse with the swtiching control assumption and also provided counterexample when the assumption does not hold.

**Weaknesses:**

1. The swtiching control assumption is kind of strong and limits the significance of the results. This paper would be much stronger with a more general sufficient condition of efficient computation of NE in zero-sum polymatrix Markov game. Currently, computation among  zero-sum polymatrix Markov game is rather unclear.

Some minor comments:

1. Check the notation of $V_{k,h}^{\pi}$ in line 161, 164 and some other places. As the text suggests, it means the cumulative reward of player $k$ after timestep $h$ but what is written refers to $V^{\pi}_{k,1}$?
2. Adding a formula might be helpful for readers to understand Assumption 2
3. Line 324: "can (be) modelled"

**Questions:**

1. Is it possible to design *decentralized* algorithm with convergence to NE in switch-control zero-sum polymatrix Markov games?
2. Zero-sum polymatrix normal-form games belong to the more general class of monotone games. Do the authors have any insights on the possiblity of generalizing monotone games to monotone Markov games and also ensure efficient computation of NE?
3. Does the set of correlated equiliria (CE) collapses to NE in zero-sum polymatrix Markov games without the swtiching control assumption?

---

> ### Author Rebuttal · Authors · 2023-08-08
>
> We would like to thank the reviewer for the time and effort they put into reviewing our work. Further, we would like to thank them for positively assessing our work. We believe that their suggestions and comments are important in improving the readability and quality of our text.
>
> >* The swtiching control assumption is kind of strong and limits the significance of the results. This paper would be much stronger with a more general sufficient condition of efficient computation of NE in zero-sum polymatrix Markov game. Currently, computation among  zero-sum polymatrix Markov game is rather unclear.
>
> > Does the set of correlated equilibria (CE) collapses to NE in zero-sum polymatrix Markov games without the switching control assumption?
>
> In the counterexample that we provide, the particular CCE we describe is also a CE. Hence the answer is no. (Please check that for the swaps $a_1\to a_2, a_2\to a_1$ in $h=1, s_1$, the value of player $1$ gets worse --- we note that swaps in $h=2$ do not change the value.)
>
> But, any transition rule that depends has the property of depending on the marginals of a correlated policy would work ensuring that the following equality holds:
> $$ P(s'|s, a, \sigma_{-k}) = P(s'|s, a, \pi^\sigma_{-k}) .$$
>
> Such a rule could be picking a controller at random at each state.
>
> Nevertheless, when the transition function depends on a product policy even of solely two players, collapse fails to hold.
>
> **Hardness of two-controller zero-sum polymatrix MGs**. Further, it is not very hard to observe that surprisingly **nonstationary** NE are PPAD hard in a zero-sum polymatrix Markov game when our assumption does not hold (rather than just *stationary* NE which we expect to be harder even with a switching controller). Please find in the rebuttal PDF the graph of the construction needed.
>
> Consider a $2$-player general-sum game $\Gamma$ with payoff matrices $(\mathbf{U}, \mathbf{V})$ for player $1,2$ accordingly. Pure strategies of players $1$ and $2$ are denoted $a_i, b_j$, accordingly, with $i\in [m]$ and $j\in[n]$. Hence, $\mathbf{U}, \mathbf{V} \in \mathbb{R}^{m\times n}$.
>
> We construct a $3$-player polymatrix zero-sum Markov game $\Gamma'$ as follows:
>
> * the time horizon of the game is $H=3$,
>
> * players $1, 2$ have the same set of available actions as players in game $\Gamma$;  $\\{ a_{i} \\}$, $\\{ b_{j} \\}$ ; the action-set of player $3$ is a singleton (dummy player),
>
> * there is an initial state $s_0$,
>
> * for every pair of actions $a_i, b_j$ of the initial game there is a state $s_{ij}$; *i.e.*, $\mathcal{S} = \{ s_{ij}, ~ij\in [m]\times [n]\}$
>
> * in state $s_{ij}$ player $1$ gets reward $U_{ij}$, player $2$ gets $V_{ij}$ and player three gets $-(U_{ij}+V_{ij})$; in $s_0$, they all get reward $0$,
> * transitions are deterministic and $P(s_{ij}|s_{0}, a_i, b_j) = 1$, while states $s_{ij}$ are absorbing.
>
> The value functions of players $1,2$ for policies in $s_0$ $\mathbf{x}:=\pi_1(s_0,h=1), \mathbf{y}:=\pi_2(s_0,h=1)$ are:
> $$
>         V_{1}(s_0) =0 + \sum_{a,b}\sum_{s_{ij}\in\mathcal{S}}x(a)y(b) P(s_{ij}|s_0, a,b)U_{ij}  =  \sum x(a_j) y(b_j) U_{ij} = \mathbf{x}^\top \mathbf{U} \mathbf{y}\\
> $$
> $$ V_2(s_0) = \mathbf{x}^\top \mathbf{V} \mathbf{y}. $$
>
> Hence, Nash equilibria of game $\Gamma$ coincide with the $\mathbf{x}, \mathbf{y}$ policies of Nash equilibria in game $\Gamma'$.
>
> >Zero-sum polymatrix normal-form games belong to the more general class of monotone games. Do the authors have any insights on the possiblity of generalizing monotone games to monotone Markov games and also ensure efficient computation of NE?
>
> This is a really interesting question for future work. In fact, we believe that our proof might be modifiable to prove the same collapse property in monotone Markov games given a similar assumption on the transition function.
>
> > Is it possible to design decentralized algorithm with convergence to NE in switch-control zero-sum polymatrix Markov games?
>
> Algorithms in [1], [2], [3] are to differing extents decentralized. Hence, it is possible.
>
>
>
>
> Thanks,
> The authors.
>
>
> ---
>
> [1] Cui, Q., Zhang, K. and Du, S., 2023, July. Breaking the curse of multiagents in a large state space: Rl in markov games with independent linear function approximation. In The Thirty Sixth Annual Conference on Learning Theory (pp. 2651-2652). PMLR.
>
> [2] Wang, Y., Liu, Q., Bai, Y. and Jin, C., 2023. Breaking the curse of multiagency: Provably efficient decentralized multi-agent rl with function approximation. arXiv preprint arXiv:2302.06606.
>
> [3] Daskalakis, C., Golowich, N. and Zhang, K., 2023, July. The complexity of markov equilibrium in stochastic games. In The Thirty Sixth Annual Conference on Learning Theory (pp. 4180-4234). PMLR.

---

> > ### Comment · Reviewer_zxYQ · 2023-08-16
> >
> > I thank the authors for the very detailed response. I have no futhur questions and think this a good paper worthy of acceptance.

---

### Official Review · Reviewer_5Tpt · 2023-07-06

**Soundness:** 3 good
**Presentation:** 2 fair
**Contribution:** 4 excellent
**Rating:** 7
**Confidence:** 4

**Summary:**

This paper considers the problem of computing approximate Nash equilibria in Markov (aka stochastic) games. It is known that such equilibria can be computed efficiently in the non-stochastic setting when the game is zero-sum polymatrix. The authors show that approximate equilibria can also be computed efficiently in zero-sum polymatrix Markov games, when the game has the switching controller property (i.e., a single, but possibly different, agent influences the transition at each step).

The result is proved by showing that the coarse-correlated equilibria (CCE) collapse to Nash equilibria in this setting. As a result, it suffices to compute a CCE, which can be done efficiently using prior work. The same approach of equilibrium collapse was also used by Cai et al. (2016) in the non-stochastic setting. The authors also show that equilibrium collapse fails to hold if the switching controller assumption is removed.

**Strengths:**

- the paper studies a very natural computational problem
- the results are obtained by a non-trivial generalization of the techniques used by Cai et al. (2016)

**Weaknesses:**

- the writing is quite sloppy in some parts

**Questions:**

Questions:

1. In Corollary 3.2 you say that you can only compute a nonstationary equilibrium. Can you comment on this? Why is that the case? Is the problem for stationary equilibria open? Are there some obstacles?

2. Your results yield a polynomial time algorithm for the case where epsilon is inverse polynomial  in the other parameters of the problem. What about the case where epsilon is inverse exponential? Is there a hardness result known for this case, or is this an open question? In other words, is it known whether an algorithm with running time poly(log(1/epsilon)) should be possible?


Further Comments:

- it would be nice to explain in the introduction why the zero-sum polymatrix setting captures both competition and coordination

- lines 539-541: it looks like something is missing here

- lines 558-559, Lemma A.2: Do you mean P_NE here? (the finite horizon case)

- lines 726-736, Proof of Theorem C.2: It looks like gamma is missing in the equations in this proof

- line 125: here you probably also want to allow H = \infty

- line 161: there seem to be some typos in the equation below this line. "h" is used both as a parameter of the quantity that is defined here, and also on the right-hand side as an index for the sum. The same applies below as well.

- line 205: the notation "A_{argctrl(s)}" is informal. The co-domain of the function cannot depend on an input of the function.

- line 216: I think the O(epsilon) notation should not be used in the definition here. Please expand and use quantifiers.

- line 245: It would be nice to add some detail about which parts of the program are linear and which are not.

- Figure 1: There are some typos in this figure. Please check that the edge labels are correct.

- line 308: There are many typos in P'_NE. The version in the appendix is correct.

- line 321: what is H? It seems like there should be no H in the infinite horizon case. Also shouldn't gamma appear in the bound somewhere? I assume that H is the expression on line 317, but this is not a parameter of the problem, so you replace the dependency on H by a dependency on 1/(1-gamma) in the bound.

- line 342-343: what is the exact dependence on epsilon in your algorithm?


Typos:
- line 59: remove "that"
- lines 72-73: sentence not finished
- line 175: remove "a an" -> "an"
- line 215: defined -> be defined
- line 289+299: uniformally -> uniformly
- line 318: sentence not finished
- line 321: proabiblity
- line 324: modelled -> be modelled

---

> ### Author Rebuttal · Authors · 2023-08-08
>
> We thank the reviewer for their positive evaluation of our work, their thorough read,ing and very diligent editing comments. Your comments are very valuable in improving the quality of our text. We will most certainly address those in our revised text. Let us try to answer your questions.
>
>
> > Questions:
> * In Corollary 3.2 you say that you can only compute a nonstationary equilibrium. Can you comment on this? Why is that the case? Is the problem for stationary equilibria open? Are there some obstacles?
>
> In general, computing a stationary Markovian CCE in a stochastic game is PPAD --- see [3]. Our setting resembles that in the sense that turn-based control is a special case of switching control. We do not know whether it is possible to get stationary Markovian policies that are NE in time polynomial in $1/\epsilon$.
>
> Further, as we observe in the rebuttal note for reviewer zxYQ, even **nonstationary** NE are hard to compute in zero-sum polymatrix games with more than 3 players and 2 controllers. We note that [1],[2] show the hardness of computtation of **stationary CCE** in two-player general-sum games.
>
> > * Your results yield a polynomial time algorithm for the case where epsilon is inverse polynomial in the other parameters of the problem. What about the case where epsilon is inverse exponential? Is there a hardness result known for this case, or is this an open question? In other words, is it known whether an algorithm with running time poly(log(1/epsilon)) should be possible?
>
>
> Thank you for this interesting question. It is reasonable that the polynomial expectation property holds for the rewards of every state [4] due to the polymatrix structure. We can answer the question:
> * for finite-horizon games where the horizon is fixed
> * for infinite-horizon games where the parameter $\gamma$ depends polynomially on the rest parameters of the game. (We truncate the horizon of the game to the effective horizon $\log(1/\epsilon)/(1-\gamma)$ and solve for a nonstationary Markov policy of the finite game.)
>
> In both these cases, we can use backwards induction along the 'ellipsoid against hope' algorithm to get a dependence polynomial in $\log(1/\epsilon)$.
>
> But, if the parameter $\gamma$ does a polynomial dependence on the rest of the parameters of the game, then we cannot really answer.
>
> > * Figure 1: There are some typos in this figure. Please check that the edge labels are correct.
> You are right. We will fix this.
>
>
> >* line 321: what is H? It seems like there should be no H in the infinite horizon case. Also shouldn't gamma appear in the bound somewhere? I assume that H is the expression on line 317, but this is not a parameter of the problem, so you replace the dependency on H by a dependency on 1/(1-gamma) in the bound.
>
> The parameter $H$ is defined as the 'effective horizon', $H = \frac{\log(1/\epsilon)}{1-\gamma}$. But it is correct that we should.
>
> > * line 342-343: what is the exact dependence on epsilon in your algorithm?
>
> The algorithm proposed by [3] has a dependence $\tilde{O}(\epsilon^{-3})$. Newer results in [1], [2] have a dependence $\tilde{O}(\epsilon^{-2})$
>
> ---
>
> Thanks,
>
> The authors
>
> ---
>
> [1] Cui, Q., Zhang, K. and Du, S., 2023, July. Breaking the curse of multiagents in a large state space: Rl in markov games with independent linear function approximation. In The Thirty Sixth Annual Conference on Learning Theory (pp. 2651-2652). PMLR.
>
> [2] Wang, Y., Liu, Q., Bai, Y. and Jin, C., 2023. Breaking the curse of multiagency: Provably efficient decentralized multi-agent rl with function approximation. arXiv preprint arXiv:2302.06606.
>
> [3] Daskalakis, C., Golowich, N. and Zhang, K., 2023, July. The complexity of markov equilibrium in stochastic games. In The Thirty Sixth Annual Conference on Learning Theory (pp. 4180-4234). PMLR.
>
> [4] Papadimitriou, C.H. and Roughgarden, T., 2008. Computing correlated equilibria in multi-player games. Journal of the ACM (JACM), 55(3), pp.1-29.

---

> > ### Comment · Reviewer_5Tpt · 2023-08-13
> >
> > Thank you for your very detailed reply. I think it would be nice to add a brief discussion about these two points to the paper. In particular, the first question could be mentioned as an open question.

---

### Official Review · Reviewer_PoXL · 2023-07-07

**Soundness:** 3 good
**Presentation:** 3 good
**Contribution:** 2 fair
**Rating:** 7
**Confidence:** 4

**Summary:**

This paper explores a specific class of Markov games called zero-sum polymatrix Markov games, a stochastic generalization of one-shot polymatrix Markov games, and shows how an ϵ-approximate Nash equilibrium can be efficiently computed. The authors show equality between the set of coarse-correlated equilibria and Nash equilibria for a subset of  these games with switching control. This implies that Nash equilibria in switching control zero-sum polymatrix Markov games can be computed efficiently. The paper also discusses open questions related to using a policy optimization algorithm to converge to an approximate Nash equilibrium. Overall, the paper provides a theoretical framework for reasoning about strategic interactions over dynamically changing networks.

**Strengths:**

The paper tackles a well-defined question, which is of interest to the community and which I would also find interesting. Overall the authors seem precise in their analysis. Writing is also often clear and and results seem reproducible.

**Weaknesses:**

The paper is not entirely self-contained (for instance when it comes to the references to Daslakakis et al’s algorithm), and lacks important intuition for the programs provided and theorem proofs. Given that the authors seems use traditional tools to obtain their results, and the paper is entirely theoretical I would expect a much more thorough explanation of theoretical result and would have liked a discussion of the algorithm introduced by Daslakakis et al. since the paper is recent and the reader might not be familiar with it.

For a purely theoretical paper, I would expect more intuition to be provided and more explanations of results. As it is, the reader has to spend its entire time decoding the results without any help from the authors. To be entirely clear, I think that the results, although they seem to rely on standard theoretical tools, are interesting and valuable to the community but the paper would be in much better shape with either experimental evaluation or with additional explanations. I remain open to changing my score and would appreciate it if the authors provide intuition and explanation of their results to this end.

**Questions:**

If polymatrix games are solvable in polytime, even in the one-shot setting, why aren’t general-sum polymatrix games solvable? Can’t I just add for any polymatrix game an additional dummy player to make the game zero-sum without changing the equilibria?

Can you build a regular game from a switching controller game by adding for each state a number of states equal to the number of players and where players get no rewards, and make the game go through these additional states for each individual state in the regular game so as to simulate more complicated transitions?

Is the set of CCE, and consequently the set of NE, in a zero-sum polymatrix Markov game convex?

There is no explanation of why the set of CCE and NE coincide, why does this happen? The proof does not give me much intuition, can you describe in a few sentences why the “collapse” happens?

I am not sure I understand Claim A1, how can we known that the correlated policy can be marginalized such that pi = pi_1 x pi_2. In general, it should be pi = pi_1|pi_2 x pi_2. What am I missing?

Can you give an inuitive explanation of the program P_{NE}. Is the program convex? I assume not, if so, is it incave/satisfy a gradient dominance condition? What is the Algorithm given by Daslakakis et al. and why does it work here?

Notes: Definition of value function (line 161): variable h overloaded on lhs and rhs

---

> ### Author Rebuttal · Authors · 2023-08-08
>
> We thank the reviewer for their challenging questions, constructive comments, and of course their time. Due to space constraints, we cannot answer as detailed as we would like to all of your questions. Please, see the global rebuttal for some further elaboration on our answers.
>
> > Solving nonzero-sum polymatrix games by adding a dummy player and making it zero sum [...].
>
> This would hint at the existence of a *polynomial time reduction*. One could even try to add multiple dummy players (not just one) to create a zero-sum polymatrix game. But, because of the established results (Chen et al. 2009), unless P=PPAD, it is impossible that any such polynomial time reduction exists. In general, one cannot hope to construct a zero-sum polymatrix game and maintain the structure of the NE.
>
> > ''*Can you build a regular game from a switching controller game by adding for each state a number of states equal to the number of players and where players get no rewards, and make the game go through these additional states for each individual state in the regular game so as to simulate more complicated transitions?*''
>
> This is a really interesting question. We are afraid that you cannot emulate any Markov game through a switching controller one without blowing up the state space.
>
> On a bright note, our proof goes through even if a controller is picked at random at every state – admittedly a less restricted case.
>
> > ''*Is the set of CCE, and consequently the set of NE, in a zero-sum polymatrix Markov game convex?*''
>
> No, please see the global rebuttal.
>
> > ''*There is no explanation of why the set of CCE and NE coincide, why does this happen? The proof does not give me much intuition, can you describe in a few sentences why the “collapse” happens?*''
>
> A very short overview of the proof is that the structure of the reward makes it so that best-responding to a correlated policy of your neighbors is no different than best-responding to what they individually play in expectation (their marginal strategies) --- exactly as in normal-form polymatrix games. This is due to the fact that players interact only in a pairwise manner and the individual utility of each player is the sum of the utilities gained in those **pairwise interactions**.
>
> Then, the extra assumption of **switching control**, makes it so that the latter phenomenon is carried over to the value function rather than just the instantaneous rewards.
>
>
> > ''*I am not sure I understand Claim A1, how can we known that the correlated policy can be marginalized such that pi = pi_1 x pi_2. In general, it should be pi = pi_1|pi_2 x pi_2. What am I missing?*''
>
> You are correct, it need not be the case that the correlated policy $\sigma$ can be marginalized to be equal to a product policy. On the contrary, we note that $\pi^\sigma$ is the product policy that results from *marginalizing* a correlated policy $\sigma$.
> I.e., for any player $k$, $\pi_k(a|s) = \sum_{\mathbf{a}' \in \mathcal{A}_{-k}} \sigma(a, \mathbf{a}'| s)$.
>
> Then, in Claim A.1 we show that when player 2 deviates from a correlated policy $\sigma$, they experience the marginal policy of player 1, $\pi_1^\sigma$, i.e. the value function is $V(\pi^\sigma_1, \cdot)$ (same goes for player 1 due to symmetry). Where $\pi^\sigma_1(a|s) = \sum_{b} \sigma(a,b|s)$ and correspondingly defined for $\pi^\sigma_2$.
>
> Note that this is not generally true even for convex-concave functions (let, alone nonconvex-concave like the value function).
> Consider the game between $x, y \in [0,1]$ with utility $f(x,y) = -x^2 + y^2$. A correlated strategy would be half of the times players jointly play $\{x=0, y=1\}$ and the rest $\{ x=1, y=0 \}$.
>
> Define $y^\sigma = 1/2 \cdot 1 + 1/2 \cdot 0$, then $f(x, y^\sigma) = - x^2 + (1/2 )^2  = -x^2 + 1/4$. While in reality, when player $x$ unilaterally deviates they experience $\frac{1}{2}f(x, 0) + \frac{1}{2} f(x,1) = - x^2 + 1/2$.
>
> > ''*Can you give an intuitive explanation of the program P_{NE}. Is the program convex? I assume not, if so, is it incave/satisfy a gradient dominance condition?*''
>
> * The first set of constraints, i.e. $w_{k,h}(s) \geq r_{k,h}(s,a,\pi_{-k}) + P(s,a,\pi_{-k}) w_{k,h+1}$, makes sure that the vectors $w_k$ are equal to the value of the **best-response** of player $k$.
> * The rest of the constraints make sure that $\pi_k$ is in the simplex and the last step of the horizon has a value equal to zero.
>
> The **objective function** minimizes the gain all players get when they unilaterally deviate. When the sum of gains (no individual gain can be negative) turns to zero, this means no player can gain anything by deviating; this is of course a NE. To see why $w_k$ holds the value of player $k$’s best-response, fix $x$ and all other $w_\ell, ~\ell\neq k$ and observe that the program coincides with the linear program for the optimal value of a single-agent MDP.
>
> Please also check the global rebuttal for further details.
>
>
> > ''*What is the Algorithm given by Daslakakis et al. and why does it work here?*''
>
> For completeness, we will include the algorithm. We initially did not include an explicit description of an algorithm in order to stress the message that **any** algorithm that computes a **stationary/nonstationary Markovian CCE** can be used as a black-box to compute a **stationary/nonstationary Markovian NE** in this class of games. This fact follows from our proof of equilibrium collapse (Theorems 3.3, C.1) -- hence, the algo. by Daskakis et al. which computes a nonstationary Markovian CCE is appropriate for our goal of computing a NE. Further, there are more recent solutions that guarantee a better convergence rate than Daskalakis et al. do, namely $\tilde{O}(\epsilon^{-2})$.
>
> Please also see the global rebuttal.
>
> ---
> Thank you,
>
> The authors

---

> > ### Author Response · Authors · 2023-08-15
> > **Further inquiries?**
> >
> > Dear reviewer,
> >
> > Since the end of the discussion period is nearing, we would like to know whether you have further questions. We would like to have the opportunity to discuss and resolve potential further inquiries.
> >
> > Thanks in advance,
> >
> > The authors

---

> > > ### Comment · Reviewer_PoXL · 2023-08-21
> > >
> > > I thank the authors for their time trying to elucidate some of these points.
> > >
> > > My earlier score was harsh, and some of the confusion was due to my misreading of the text I believe, and I am increasing my score in light of the new answers. I support the paper's acceptance, and I hope the author's can improve the writing of the paper for the camera-ready version.

---

### Author Rebuttal · Authors · 2023-08-08

The authors would like to thank the reviewers for their constructive comments, their positive assessment, and their valuable time. Surely, the reviews will significantly improve the readability of our work. We apologize for the typos and, at times, parts of our writing might have been confusing.

Some comments are in order:

**On the mathematical programs.**
It seems that most reviewers were concerned with the lack of a note of an intuitive explanation of the mathematical programs. We understand that this was an oversight on our part and we will complement our text with such a note. We first note that the program is nonlinear due to the multiplication of variables $\pi_i$ and $v_i$ with each other in the term $P(s, a, \pi_{contrl.} ) v$. Now, allow us to provide a very short note on the intuition behind the program.

The first set of constraints makes sure that the vectors $w_k$ are equal to the value of the best response of player $k$. The objective function minimizes the gain all players get when they unilaterally deviate. When the sum of gains (no individual gain can be negative) turns to zero, this means no player can gain anything by deviating; this is of course a NE.

To see why $w_k$ holds the value of player $k$’s best-response, fix $\pi$ and all other $w_\ell, ~\ell\neq k$ and observe that the program coincides with the linear program for the optimal value of a single-agent MDP .

*Convexity/Nonconvexity of the program*: The objective function for the special case of zero-sum games is convex (linear to be more precise). Yet, the constraint set is nonconvex. The program is never solved; it is a tool for the proof and we are not actually concerned with those properties. The property that we care about is the fact that global minima correspond to NE and vice versa. It is not readily obvious to us how the gradient domination property should be defined in terms of a nonconvex constraint set.


**The algorithm.**
It is our understanding that the lack of a description of an explicit algorithm is a mistake and we will add an appropriate algorithm from the existing literature in the appendix of our reviewed text. The reason we omitted it is due to the fact that we wanted to stress the message that any algorithm that computes a **stationary/nonstationary Markovian CCE** can be used as a *black-box* to retrieve **stationary/nonstationary Markovian NE** for this particular setting of games. Hence, algorithms found in [3],[4],[5] all can be used. A similar omission of particular algorithms was made in papers like (Cai et al. 2016), (Even-Dar et al. 2009) when proving that any no-regret algorithm suffices in order to get a NE in their settings.


**The switching control assumption.** We note that our proof with virtually no change for any transition rules that depend on the marginal distribution of a correlated policy, i.e.,  $ P_h(s, a, \sigma_{-k}) = P_h(s, a, \pi^{\sigma}_{-k}).$ Also, it is worth taking a look at the observation we make in the rebuttal for reviewer zxYQ that zero-sum polymatrix Markov games with 3 players and 2 controllers are PPAD hard.


**Nonconvexity of the CCE set.** We create a very simple zero-sum polymatrix switching controller MG using a simple MDP and adding a dummy second agent.

* We first consider a single player (player $1$) facing a deterministic MDP.
* The optimal policies in that MDP do not form a convex set.
* We add a dummy player who gets exactly the opposite of the initial agent. The dummy player has a single action at their disposal and does not affect the transitions or the rewards.
* Player $1$ remains the controller in every state.
* CCE and NE coincide in this game (the action space of player $2$ is a singleton).
* Nash equilibria coincide with the optimal policies of the MDP which do not form a convex set.

We define the MDP,
* There are $5$ states and the game begins at state $s_1$.
* In the MDP, the agent can choose to go $\texttt{UP}:a_1$ or $\texttt{RIGHT}:a_2$ at every state.
* In $s_1$

    * $a_1$ gets a reward of $2$,
    * $a_2$ gets $1$,

  In $s_2$,

    * $a_1$ gets a reward of $1$,
    * $a_2$ gets $0$,
* States $\{s_3, s_4, s_5\}$ are absorbing.

The value function of player $1$ is,
\begin{align}
    V(\pi) = 2\pi(a_1|s_1) + \pi(a_2|s_1) + \pi(a_2|s_1) \pi(a_1|s_2)
\end{align}

Two optimal strategies are $\pi', \pi''$:
\begin{align}
    \begin{cases}
        \pi'(a_1|s_1) = 1, \pi'(a_2|s_1) = 0\\\\
        \pi'(a_1|s_2) = 1, \pi'(a_2|s_2) = 0
    \end{cases}
    &
    \begin{cases}
        \pi''(a_1|s_1) = 0, \pi''(a_2|s_1) = 1\\\\
        \pi''(a_1|s_2) = 0, \pi''(a_2|s_2) = 1
    \end{cases}
\end{align}


$V(\pi')= V(\pi'') = 2$.

\begin{align}
    \begin{cases}
        \bar{\pi}(a_1|s_1) = 1/2,&  \bar{\pi}(a_2|s_1) = 1/2\\\\
        \bar{\pi}(a_1|s_1) = 0,  &\bar{\pi}(a_2|s_1) = 1 \\\\
    \end{cases}
\end{align}

But the value of $\bar{\pi}$ (a convex combination of optimal policies) is not optimal,
\begin{align}
    V(\bar{\pi}) = 2\cdot1/2 + 1/2 + 1/2 \cdot 1/2 = 7/4.
\end{align}

---

Thank you,

The authors

---

[1] Puterman Markov decision processes: discrete stochastic dynamic programming 2014

[2] Neu, G. and Pike-Burke, C., 2020. A unifying view of optimism in episodic reinforcement learning. Advances in Neural Information Processing Systems, 33, pp.1392-1403.

[3] Cui, Q., Zhang, K. and Du, S., 2023, July. Breaking the curse of multiagents in a large state space: Rl in markov games with independent linear function approximation. In The Thirty Sixth Annual Conference on Learning Theory (pp. 2651-2652). PMLR.

[4] Wang, Y., Liu, Q., Bai, Y. and Jin, C., 2023. Breaking the curse of multiagency: Provably efficient decentralized multi-agent rl with function approximation. arXiv preprint arXiv:2302.06606.

[5] Daskalakis, C., Golowich, N. and Zhang, K., 2023, July. The complexity of markov equilibrium in stochastic games. In The Thirty Sixth Annual Conference on Learning Theory (pp. 4180-4234). PMLR.

---

### Decision · Program_Chairs · 2023-09-21

**Decision:**

Accept (poster)

**Comment:**

This paper identifies a new type of multi-player general-sum Markov game, namely zero-sum polymatrix Markov Games with switching controllers, under which a Nash Equilibrium can be computed in polynomial time by the "equilibrium collapse" property. All reviewers are positive about the contributions. Therefore, I recommend acceptance.